# C-NHEJ without indels is robust and requires synergistic function of distinct XLF domains

Ragini Bhargava[1,2], Manbir Sandhu[2,3], Sanychen Muk [3], Gabriella Lee[1], Nagarajan Vaidehi[2,3] & Jeremy M. Stark [1,2]

To investigate the fidelity of canonical non-homologous end joining (C-NHEJ), we developed an assay to detect EJ between distal ends of two Cas9-induced chromosomal breaks that are joined without causing insertion/deletion mutations (indels). Here we find that such EJ requires several core C-NHEJ factors, including XLF. Using variants of this assay, we find that C-NHEJ is required for EJ events that use 1–2, but not ≥3, nucleotides of terminal microhomology. We also investigated XLF residues required for EJ without indels, finding that one of two binding domains is essential (L115 or C-terminal lysines that bind XRCC4 and KU/DNA, respectively), and that disruption of one of these domains sensitizes XLF to mutations that affect its dimer interface, which we examined with molecular dynamic simulations. Thus, C-NHEJ, including synergistic function of distinct XLF domains, is required for EJ of chromosomal breaks without indels.

[1] Department of Cancer Genetics and Epigenetics, Beckman Research Institute of the City of Hope, 1500 E Duarte Rd., Duarte, CA 91010, USA. [2] Irell and Manella Graduate School of Biological Sciences, Beckman Research Institute of the City of Hope, 1500 E Duarte Rd., Duarte, CA 91010, USA. [3] Department of Molecular Immunology, Beckman Research Institute of the City of Hope, 1500 E Duarte Rd., Duarte, CA 91010, USA. These authors contributed equally: Manbir Sandhu, Sanychen Muk. Correspondence and requests for materials should be addressed to J.M.S. (email: jstark@coh.org)

End-joining (EJ) repair of chromosomal DNA double-strand breaks (DSBs) is critical for genome maintenance and cellular resistance to clastogens, but also can generate oncogenic chromosomal rearrangements. Characterizing the factors and pathways that influence the fidelity and efficiency of EJ is important for understanding cancer etiology and response to clastogenic therapeutics. A major pathway of such repair is canonical/classical non-homologous end joining (C-NHEJ), which involves the core factors KU (KU70/80), DNA-PKcs, XRCC4, XLF, and DNA ligase IV (LIG4)[1,2]. However, while loss of C-NHEJ factors causes substantial clastogen sensitivity, the Alternative-EJ (Alt-EJ) pathway provides some redundancy[3–5].

Such partial redundancy between C-NHEJ and Alt-EJ has been observed for several EJ events, including EJ between two endonuclease-generated chromosomal breaks and AID-induced breaks during class switch recombination[6–9]. Namely, in the absence of C-NHEJ these events are readily detectable, albeit at a reduced frequency[6–9]. In addition, the requirement for C-NHEJ during V(D)J recombination is specific to events using the full-length RAG recombinase[3]. Specifically, deleting a C-terminal domain of RAG can rescue the requirement for C-NHEJ factors (e.g., XRCC4) during V(D)J recombination, such that Alt-EJ is proficient for V(D)J recombination in this context[3]. However, without C-NHEJ, repair junctions often show a higher frequency of microhomology[3–8]. Thus, C-NHEJ and Alt-EJ appear to mediate distinct EJ repair outcomes, although the precise EJ events that distinguish these pathways have remained poorly understood.

Furthermore, the role of XLF during DSB repair is particularly complex. While XLF can substantially promote the activity of the XRCC4-LIG4 complex in vitro, is important for clastogen resistance, and promotes V(D)J recombination in plasmid substrates, this factor is not required for chromosomal V(D)J recombination in lymphocytes, unless in the context of loss of other DNA damage response factors[10–16]. XLF forms a homodimer, and has at least two main binding interfaces: a globular head domain that interacts with XRCC4; and a C-terminal domain that interacts with both the KU heterodimer and DNA[17–20]. However, the requirement for these binding interfaces for C-NHEJ is unclear, particularly for the XLF-XRCC4 interaction. Namely, an XLF mutant that disrupts the XRCC4 interaction (L115A) does not obviously affect XLF function, except in the context of ATM-deficiency, which is consistent with a greater requirement for C-NHEJ in ATM-deficient cells[12,17,21,22]. Accordingly, we have sought to identify a chromosomal break repair outcome that requires C-NHEJ, and subsequently define the role of distinct domains of XLF for such repair.

## Results

**Distal EJ without indels is robust and requires C-NHEJ.** We sought to develop a chromosomal DSB reporter assay that is specific for C-NHEJ vs. Alt-EJ. Considering distinctions between these pathways, EJ events in C-NHEJ-deficient cells often show an increase in microhomology at the repair junction[3–7]. Accordingly, C-NHEJ appears to be required to join DNA ends that are not stabilized by an annealing intermediate. Thus, we posited that a reporter assay that specifically measures EJ of blunt DSBs without causing insertion/deletion mutations (indels) would be specific for C-NHEJ. For this approach, we need to examine EJ between two DSBs (i.e., distal EJ), since repair of a single DSB by EJ without indels restores the original sequence, and hence is not distinct from a site that was never cleaved.

Accordingly, we developed a green fluorescent protein (GFP)-based reporter to exclusively detect distal EJ without indels (Fig. 1a, EJ7-GFP). We split the GFP coding sequence at the GGC

codon for glycine 67, which is a residue critical for fluorescence[23], by inserting a 46 nucleotide (nt) spacer between the first two bases (GG) and the final base (C). Two single guide RNAs (sgRNAs), 7a and 7b, target Cas9-induced DSBs to excise the 46-nt spacer. Since Cas9 predominantly induces blunt DSBs[24], we posited that EJ between the distal DSBs without indels would restore the GGC codon. To clarify, while this repair event deletes the 46-nt spacer between the DSBs, there are no further changes to the DSB ends, and hence is distal EJ without indels. We integrated the EJ7-GFP reporter into the *Pim1* locus of wild-type (WT) mouse embryonic stem cells (mESCs), and found that co-expression of Cas9 with the 7a and 7b sgRNAs induced GFP+ cells at a frequency of 50%, normalized to transfection efficiency (Fig. 1b, Supplementary Fig. 1). We confirmed the expected repair product in GFP+sorted cells by PCR amplification and sequencing analysis (Fig. 1a). Thus, distal EJ without causing indels is markedly robust.

To examine the influence of C-NHEJ, we integrated EJ7-GFP into the *Pim1* locus in *Xrcc4−/−*, *Xlf−/−*, and *Ku70−/−*[10,25,26] mESCs, expressed Cas9 with the 7a and 7b sgRNAs, along with a complementation vector or control empty vector (EV). Subsequently, we determined the frequency of GFP+ cells from these transfections. We found that loss of XRCC4, XLF, and KU70, each caused a >175-fold decrease in the frequency of GFP+ cells, which could be recovered to near WT levels by co-expressing the respective complementation vector (Fig. 1b). These results indicate that the C-NHEJ factors XRCC4, KU70, and XLF are each required for EJ between two DSBs without causing indels. A corollary of this finding is that the EJ7-GFP reporter assay is highly specific for C-NHEJ.

We also tested the EJ7-GFP reporter in human cells. We chromosomally integrated this reporter into human U2OS cells using the FLP/FRT recombination system[27] and found that co-expression of Cas9 with the 7a and 7b sgRNAs induced GFP+ cells (54%, Fig. 2a). We also examined a set of C-NHEJ-deficient patient-derived fibroblast lines: 2BN cells with a frameshift mutation in *XLF*[11,28], and 411BR cells with hypomorphic mutations in *LIG4* (i.e., three amino-acid substitutions that diminish LIG4 adenylation and ligation activities)[29]. For these experiments, we introduced EJ7-GFP as an extrachromosomal plasmid that was co-transfected with Cas9 and the respective sgRNAs, as well as the respective complementation vector or control EV. We validated this extrachromosomal plasmid approach using mESCs (Supplementary Fig. 2a). From the human cell experiments, expressing the respective complementation vector caused a marked increase in GFP+ cells: 9-fold for human XLF expression in 2BN cells, and 5.4-fold for expression of human LIG4 in 411BR cells (Fig. 2b). Using immunoblotting, we confirmed that XLF is both undetectable in the 2BN cell line, and is expressed by our complementation vector (Fig. 2b). We also confirmed that the 411BR cell line retains LIG4 expression, and found that LIG4 levels are not affected by our complementation vector, likely due to only a fraction of the cells being transfected (Fig. 2b). Thus, distal EJ without causing indels in human cells is robust and dependent on C-NHEJ.

These findings with XLF were somewhat unexpected, since this factor is not required for V(D)J recombination in lymphocytes, nor for the capacity of the XRCC4-LIG4 complex to ligate blunt ends in vitro[10–16,30]. Furthermore, several studies have demonstrated substantial functional redundancy between XLF and other factors[10–16]. Indeed, combined loss of PAXX and XLF causes synthetic lethality during embryonic development[31], and defects in V(D)J recombination[16,32–35]. Thus, we examined the influence of PAXX on distal EJ without indels, by generating *Paxx−/−* mESCs, using previously described Cas9/sgRNAs[16] (Fig. 2c). While we were unable to detect PAXX protein in WT mESCs by

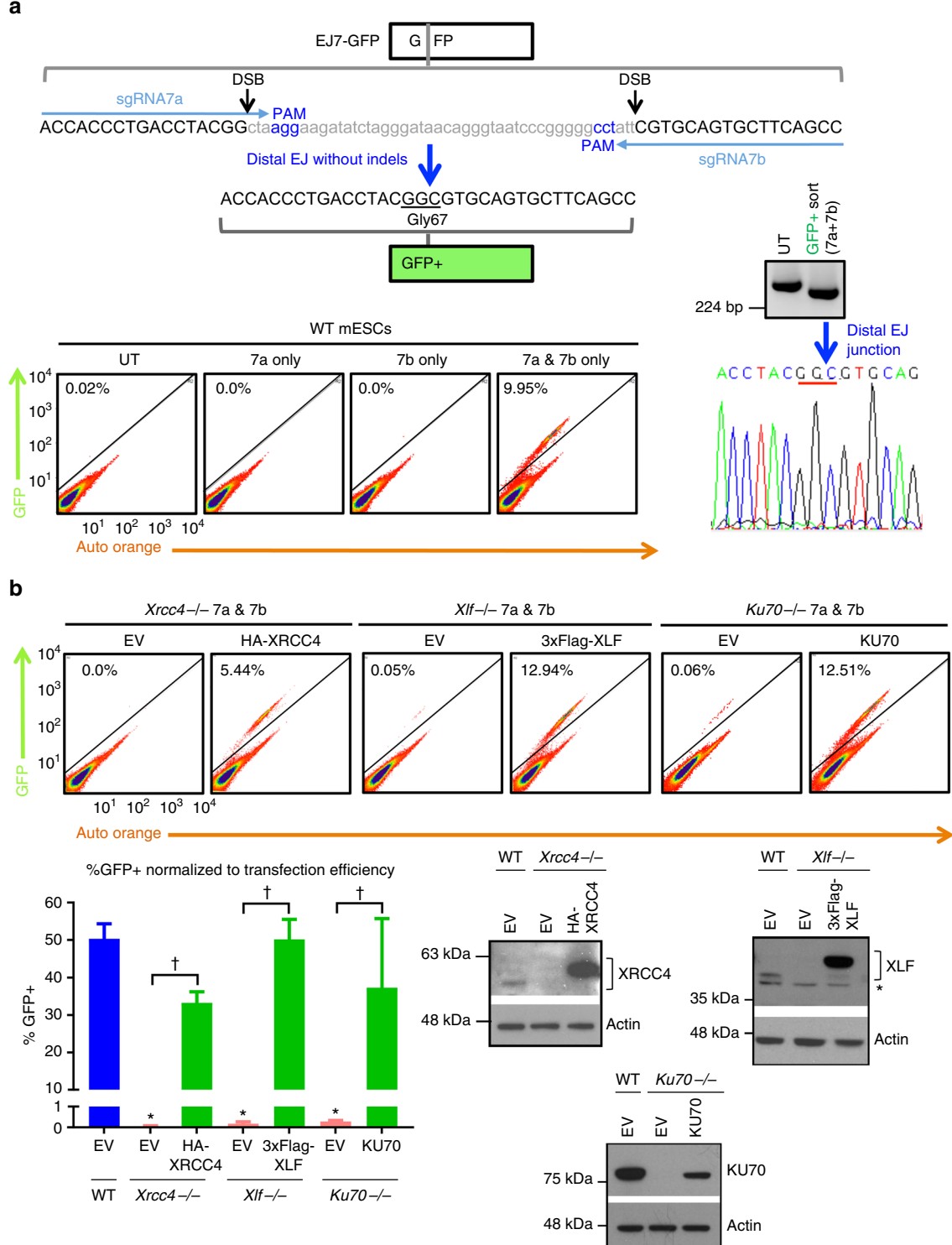

**Fig. 1** Distal end joining (EJ) without causing indels requires C-NHEJ. **a** Shown is the EJ7-GFP reporter assay for distal EJ without indels. Representative flow cytometry plots are shown for WT mESCs with EJ7-GFP integrated into the *Pim1* locus, which were untransfected (UN), or transfected with expression plasmids for Cas9 and sgRNAs (7a, 7b, or both). Shown is the amplification product and chromatogram from GFP+ WT mESCs to confirm the expected repair product. **b** Distal EJ without indels requires XRCC4, XLF, and KU70. WT, *Xrcc4−/−*, *Xlf−/−*, and *Ku70−/−* mESCs with EJ7-GFP integrated into the *Pim1* locus were transfected with expression vectors for Cas9 and the 7a and 7b sgRNAs, in the presence of a control empty vector (EV) or a complementation vector. Shown are representative flow cytometry plots, as well as the frequency of GFP+ cells for these transfections, normalized to transfection efficiency. $N = 6$, error bars represent standard deviation (s.d.). *$p < 0.0001$, mutant cell lines vs. WT using an unpaired *t*-test, with the Holm-Sidak correction (for multiple comparisons). †$p \leq 0.0006$, EV vs. complemented compared using an unpaired, two-tailed *t*-test. Also shown are immunoblots confirming expression of XRCC4, XLF, and KU70

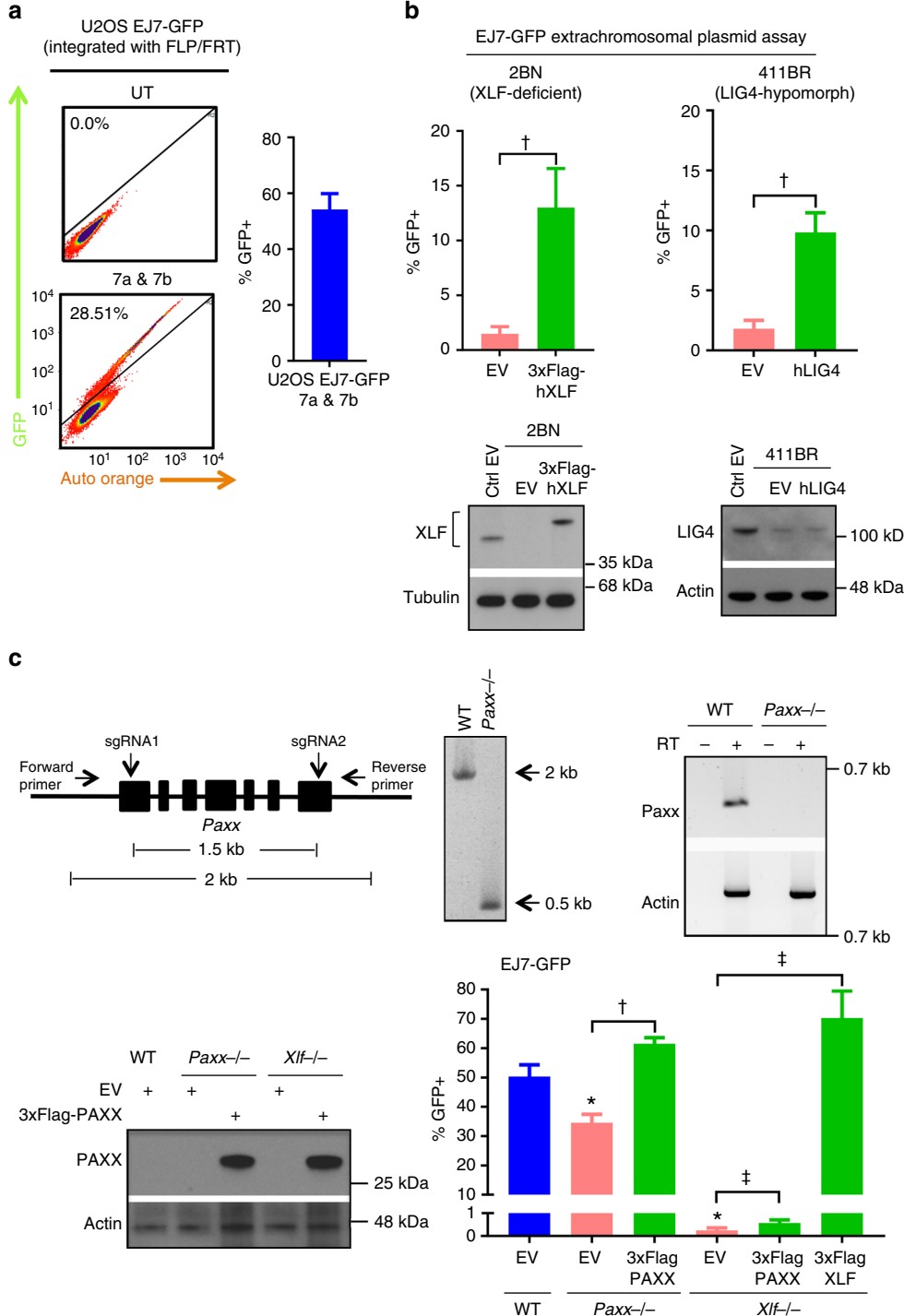

**Fig. 2** Distal EJ without indels is mediated by C-NHEJ in human cells; PAXX has a modest role on such EJ. **a** EJ7-GFP in U2OS. Shown are representative flow cytometry plots, and the frequency of GFP+ cells normalized to transfection efficiency, for the EJ7-GFP reporter integrated into U2OS cells, which were transfected with expression vectors for Cas9 and the 7a and 7b sgRNAs. N = 5, and error bar indicates s.d. **b** EJ7-GFP extrachromosomal plasmid assay in human fibroblasts. XLF-deficient (2BN) and LIG4 hypomorph (411BR) cells were transfected with EJ7-GFP, along with expression vectors for Cas9 and the 7a and 7b sgRNAs. Also included was a control (EV) or human (h) complementing vectors: 3xFlag-hXLF or hLIG4. Shown is the frequency of GFP+ cells normalized to transfection efficiency. N = 6, error bars indicate s.d. †p < 0.0001, EV vs. complemented using an unpaired, two-tailed t-test. Shown are immunoblots for XLF and LIG4. The relative migration of 3xFlag-hXLF vs. endogenous XLF is consistent with the difference in molecular weight. **c** PAXX has a modest influence on EJ7-GFP. Shown is a diagram for *Paxx* gene disruption, with PCR and RT-PCR analysis for loss of the *Paxx* gene and transcript, respectively. Shown is the frequency of GFP+ cells normalized to transfection efficiency, for WT, *Paxx−/−*, and *Xlf−/−* mESCs with EJ7-GFP, and transfected with expression vectors for Cas9 and the 7a and 7b sgRNAs, with either EV or complementation vector. N = 6, error bars indicate s.d. Shown are immunoblots for PAXX expression. *p < 0.0001, WT vs. mutants; ‡p ≤ 0.0008, EV vs. complemented, both using an unpaired t-test with the Holm-Sidak correction. †p < 0.0001, EV vs. complemented using an unpaired, two-tailed t-test

immunoblotting with an antibody raised against human PAXX, we confirmed that mESCs express *Paxx* RNA using reverse transcription-PCR (RT-PCR), and that this signal was lost in *Paxx*−/− mESCs (Fig. 2c). We integrated the EJ7-GFP reporter into the *Pim1* locus of *Paxx*−/− mESCs, and performed the reporter assay in parallel with *Xlf*−/− mESCs. We also included a mouse PAXX expression vector, or EV (Fig. 2c). We found that PAXX deficiency caused a significant but modest (1.5-fold)

decrease in distal EJ without indels, which could be restored by expressing PAXX (Fig. 2c). While loss of PAXX did not cause a dramatic effect on this EJ event, we nevertheless considered the possibility that PAXX expression might suppress the EJ defect in *Xlf*−/− cells. However, we found that PAXX expression failed to rescue this EJ event in *Xlf*−/− cells (Fig. 2c). Although, we were unable to quantify the fold-effect on PAXX levels in these experiments, since we cannot detect endogenous mouse PAXX by

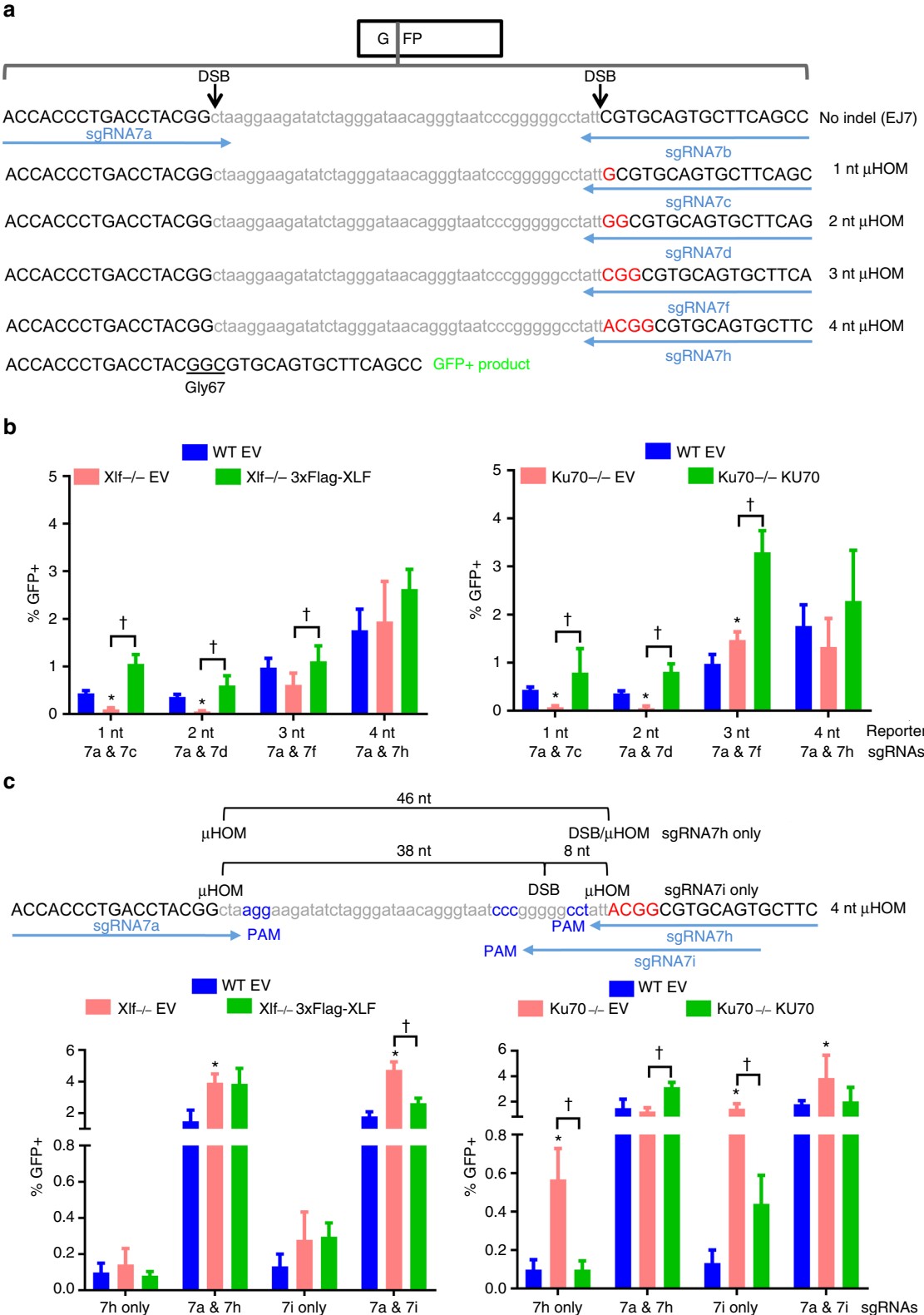

immunoblotting. In summary, PAXX plays a modest role in distal EJ without indels, and expression of PAXX cannot compensate for loss of XLF for this EJ event. These findings indicate that XLF and PAXX are not redundant for distal EJ without indels.

**Influence of KU70 and XLF on EJ with microhomology.** Using the EJ7-GFP platform, we also examined EJ events that use microhomology to cause deletion mutations. Defining the mechanism of such EJ is important for understanding the etiology of chromosomal rearrangements with microhomology[36], for which the role of C-NHEJ has remained controversial[37]. We modified the EJ7-GFP reporter by inserting 1–4 nts of terminal microhomology onto the distal end of the 3′ DSB (Fig. 3a, designated to represent the length of microhomology added as 1-, 2-, 3-, and 4-nt μHOM reporters). Due to these nucleotide additions, each of these derivatives of EJ7-GFP uses a different sgRNA to induce the 3′ DSB, but all use the 7a sgRNA to target the 5′ DSB (Fig. 3a). Following induction of these DSBs, only distal EJ events that use the respective terminal microhomology to form the corresponding deletion mutation can restore the glycine 67 codon of GFP (Fig. 3a).

We first compared the relative frequency of these EJ events, with each reporter integrated into the *Pim1* locus of WT mESCs. In comparing the different reporters, we examined both the % GFP+ cells, as well as the frequency of total distal EJ between the two Cas9-induced DSBs (i.e., excision of the spacer sequence), using PCR analysis (Supplementary Fig. 1). For EJ7-GFP and the 4-nt μHOM reporter, expression of Cas9 and the respective sgRNAs induced similar frequencies of total distal EJ, whereas the %GFP+ frequency was 12-fold lower for the 4-nt μHOM event. For the 1-, 2-, and 3-nt μHOM reporter cell lines, the frequency of total distal EJ was lower than for EJ7-GFP (6-, 25-, and 4.6-fold, respectively), however the %GFP+ frequency was reduced to a much greater extent (57-, 80-, and 32-fold, respectively). Thus, EJ events using terminal microhomology appear less frequent than EJ without indels.

We then examined the role of C-NHEJ in mediating these EJ events. We integrated each reporter into the *Pim1* locus in *Xlf* −/− and *Ku70*−/− mESCs, and transfected these cell lines with expression vectors for the associated sgRNA pairs with Cas9, along with the respective complementation vector or control EV. From these experiments, we found that loss of XLF or KU70 each resulted in a marked reduction in distal EJ events using 1 or 2 nts of microhomology, compared to WT mESCs (Figs. 3b, 1- and 2-nt μHOM reporters), which could be restored by the respective complementation vector (Fig. 3b). In contrast, distal EJ events using 3 nts of microhomology were only modestly affected by XLF or KU70 loss, and events using 4 nts of microhomology were unaffected by loss of these factors (Fig. 3b, 3- and 4-nt μHOM reporters, respectively). We found similar results using extrachromosomal reporters in *Xlf*−/− and *Ku70*−/− mESCs, and with chromosomal reporters in *Xrcc4*−/− cells (Supplementary Fig. 2a, b). These findings indicate that C-NHEJ is critical for EJ

events that use 1–2 nts, but not ≥3nts of terminal microhomology.

We next wondered how the position of microhomology relative to the edge of the DSB might affect the influence of KU70 and XLF on such EJ. For this, we designed a panel of sgRNAs to vary the DSB/microhomology distance for the 4-nt μHOM reporter. For one, we expressed an sgRNA (7 h only) to induce a DSB that is 46 nts from the 5′ microhomology, and at the edge of the 3′ microhomology (Fig. 3c). From this experiment, we found that WT cells showed a very low frequency of GFP+ cells, which was unaffected by XLF loss, however KU70 loss caused a marked increase in this repair event (Fig. 3c, 5.7-fold). We found similar results for experiments with a different sgRNA (7i only), which induces a DSB that is 8 nts from the 3′ microhomology, and 38 nts from the 5′ microhomology (Fig. 3c, 15-fold increase in *Ku70* −/− vs. WT). Thus, a long DSB/microhomology distance (e.g., 38 nts) creates a barrier to microhomology use that is dependent on KU70, but not on XLF.

We then combined the 7a and 7i sgRNAs to generate two DSBs in the 4-nt μHOM reporter, which are at the edge of the 5′ microhomology, and 8 nt from the 3′ microhomology, respectively. This sgRNA pair (7a and 7i) induced GFP+ cells and total distal EJ at similar frequencies as the sgRNA pair that induces both DSBs at the edge of the microhomology (7a and 7 h) (Fig. 3c, Supplementary Fig. 1). Using the 7a and 7i sgRNAs, we found loss of XLF and KU70 caused a modest increase in these events (Fig. 3c, 3.2- and 2.6-fold, respectively). These findings indicate that embedding 4 nts of microhomology ≤ 8 nts is not a substantial barrier for use of the microhomology, although these events are nevertheless suppressed by KU70 and XLF.

**Influence of CtIP and POLQ on EJ.** We also examined the role of other factors in these EJ events, to provide a contrast with C-NHEJ. Specifically, we examined the end resection factor CtIP and DNA polymerase theta (POLQ), both of which promote Alt-EJ[7,38,39]. We examined CtIP using siRNA, and POLQ with a previously described *Polq*−/− mESC line and expression vector for Flag-POLQ[40] (Fig. 4a, b). While detecting endogenous POLQ by immunoblotting is unfeasible, we confirmed the *Polq*−/− genotype (see Methods). We found that CtIP depletion and POLQ deficiency each caused a significant reduction in EJ using 4 nts of microhomology that is embedded 8 nts from the 3′ DSB (Fig. 4a, b, 7a and 7i sgRNAs, 5′ DSB at the edge of the microhomology). In contrast, CtIP depletion caused a modest increase in EJ involving 1–2 nts of terminal microhomology, and EJ without indels (Fig. 4a). Similarly, POLQ deficiency caused a modest increase in EJ using 1 nt of terminal microhomology, and has no effect on EJ without indels or using 2 nts of terminal microhomology (Fig. 4b). Regarding events with 3–4 nts of terminal microhomology (i.e., at the edge of both DSBs), CtIP depletion had no effect on these events (Fig. 4a), whereas POLQ had a modest effect (Fig. 4b). Specifically, EJ using 3 nts of terminal microhomology was reduced in *Polq*−/− vs. WT but not

**Fig. 3** C-NHEJ is required for EJ involving 1–2, but not ≥3, nucleotides of terminal microhomology. **a** Shown are derivatives of the EJ7-GFP reporter with increasing amounts of terminal microhomology (μHOM) downstream of the 3′ DSB (red, uppercase), which also necessitates each reporter to use a different 3′ sgRNA. Use of the microhomology to form the associated deletion mutation is required to restore the GFP coding sequence. **b** Influence of XLF and KU70 on EJ using terminal microhomology. The microhomology EJ reporters in (**a**) were integrated into the *Pim1* locus of WT, *Xlf*−/−, and *Ku70*−/− mESCs, and were transfected with sgRNA/Cas9 plasmids, along with control (EV) or complementation vector. Shown is the frequency of GFP+ cells normalized to transfection efficiency. *N* = 6, error bars indicate s.d. *$p$ < 0.004, *Xlf*−/− and *Ku70*−/− compared to WT using an unpaired *t*-test with the Holm-Sidak correction. †$p$ < 0.02, control EV vs. complemented using an unpaired, two-tailed *t*-test. **c** KU70 suppresses EJ with embedded microhomology. Shown is the 4-nt microhomology reporter in (**a**) with sgRNAs that generate DSBs positioned at varying distances from the 5′ and 3′ microhomology, which were used for transfections as in (**b**). Shown are GFP+ cells normalized to transfection efficiency. *N* = 6, error bars indicate s.d. *$p$ < 0.0003, WT vs. mutant cell line using an unpaired *t*-test with the Holm-Sidak correction. †$p$ ≤ 0.0002, control EV vs. complemented using an unpaired, two-tailed *t*-test

increased by POLQ expression, while EJ using 4 nts of terminal microhomology was not statistically lower than WT, but nevertheless POLQ expression caused a significant increase in these events (Fig. 4b). Altogether, these findings indicate that CtIP and POLQ are particularly important for EJ with 4 nts of microhomology that is embedded from the edge of the DSB.

**Influence of distinct XLF domains on EJ**. Given that XLF is required for the EJ event measured by EJ7-GFP, we used this reporter to characterize XLF functional domains (Fig. 5a)[17]. We examined two known binding interfaces of XLF: L115, which is important for interaction with XRCC4[17,18], and a C-terminal lysine-rich region, which is important for interaction with KU and/or DNA[17–19]. Since these residues are conserved between mouse and human XLF[17], and because we performed experiments in mESCs, we used mouse XLF for this analysis. First, we confirmed the importance of these residues in forming protein complexes by co-immunoprecipitation (Fig. 5a). We found that a mutant with four C-terminal lysines changed to alanine (4KA: K284A; K286A; K288A; and K289A) failed to form a complex with KU70, but was proficient at forming a complex with XRCC4 (Fig. 5a). Conversely, the L115A mutant was deficient for forming a complex with XRCC4, but retained complex formation with KU70 (Fig. 5a). Finally, the L115A/4KA double mutant failed to

form a complex with either XRCC4 or KU70 (Supplementary Fig. 2c).

We also examined two residues in the coiled-coil domain of the XLF dimer: K160 and R178. K160 has been suggested to form a salt bridge between the two XLF monomers (i.e., K160/D161')[20]. We examined a K160D mutation, which would likely disrupt this salt bridge, and found that the K160D mutant was not expressed as well as WT, yet retained the ability to form a complex with KU70 and XRCC4 (Fig. 5a). Mutation of R178 has been shown to disrupt the ability for XLF to promote XRCC4-LIG4 ligase activity, without disrupting the association of XLF with XRCC4[17]. Thus, we posited that mutation of this residue might also affect XLF function, which we changed to glutamine (R178Q), since this mutation was found in a colorectal cancer sample[41,42]. We found that R178Q complex formation with KU70 and XRCC4 was indistinguishable from WT (Fig. 5a). Furthermore, double mutations of the coiled-coil residues with L115A (L115A/K160D and L115A/R178Q) or 4KA (K160D/4KA and R178Q/4KA) did not affect the ability for XLF to form a complex with KU70 or XRCC4, respectively (Fig. 5a).

We examined the effect of these mutations on XLF function for distal EJ without indels. We performed the EJ7-GFP reporter assay in *Xlf*−/− mESCs along with 3xFlag-tagged XLF (WT or mutant) expression vectors, or control EV. We found that the

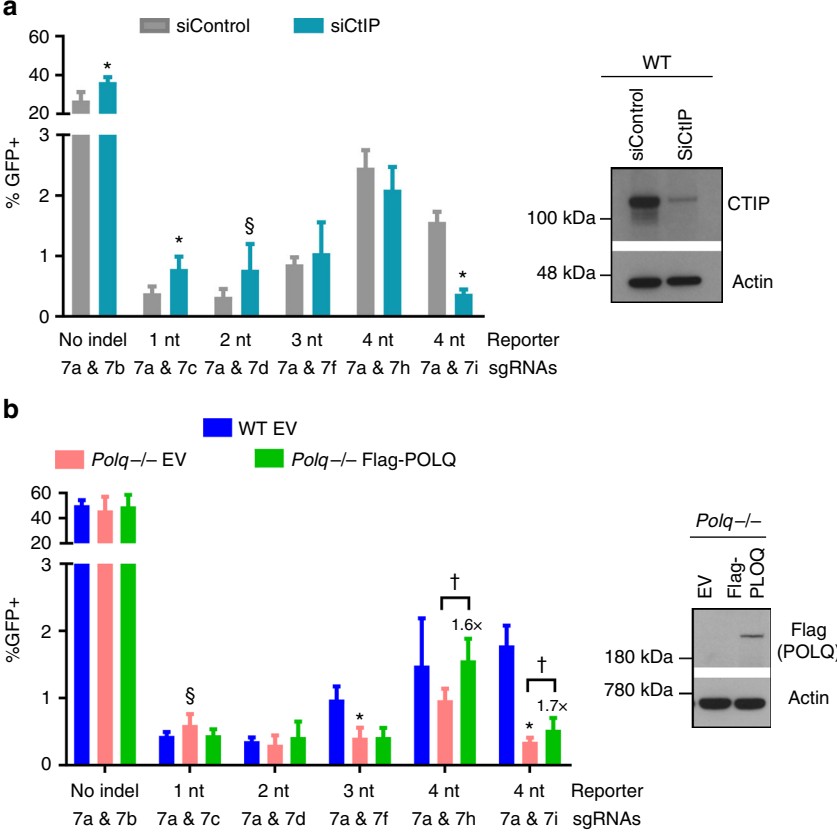

**Fig. 4** CtIP and POLQ promote EJ events using 4 nts of embedded microhomology. **a** WT mESCs with the chromosomally integrated EJ7 and microhomology reporters as in Figs. 1a and 3a were transfected with the respective sgRNA/Cas9 plasmids in the presence of either siControl or siCtIP. Shown is the frequency of the GFP+ cells normalized to transfection efficiency. N = 6. Error bars indicate s.d. *p < 0.005, siControl vs. siCtIP using an unpaired t-test with the Holm-Sidak correction. §p < 0.05, but >0.05 when adjusted for multiple comparisons. Also shown is immunoblotting analysis confirming CTIP depletion in WT mESCs. **b** The EJ7 and microhomology reporters as in Fig. 3a were integrated into *Polq*−/− mESCs, and were transfected with sgRNA/Cas9 plasmids along with a control (EV) or complementation vector. Shown is the frequency of GFP+ cells normalized to transfection efficiency. The frequencies for WT mESCs are the same as in Figs. 1b and 3b. N ≥ 6 for *Polq*−/− mESCs and N = 6 for WT mESCs. Error bars indicate s.d. *p < 0.002, WT vs. *Polq*−/− mESCs using an unpaired t-test with the Holm-Sidak correction. §p < 0.05, but >0.05 when adjusted for multiple comparisons. †p < 0.01, control EV vs. complemented using an unpaired, two-tailed t-test. Also shown is immunoblotting analysis confirming expression of the complementation vector

L115A, K160D, or 4KA mutants each showed a reduction in EJ function compared to WT, although none of these mutations abolished such EJ (Fig. 5b). In contrast, each of the double mutants (L115A/4KA, L115A/K160D, or K160D/4KA) showed nearly a complete loss in EJ function similar to a truncated (Δ1–49 amino acids) protein (Fig. 5b, >60-fold). However, the

double mutants retained residual activity above background, reflecting the wide dynamic range of this assay (Fig. 1b), compared to other assays, such as co-IP. We also found that R178Q alone caused a very minor (1.3-fold) decrease in EJ function, whereas L115A/R178Q or R178Q/4KA caused a synergistic reduction in EJ function, compared to the single

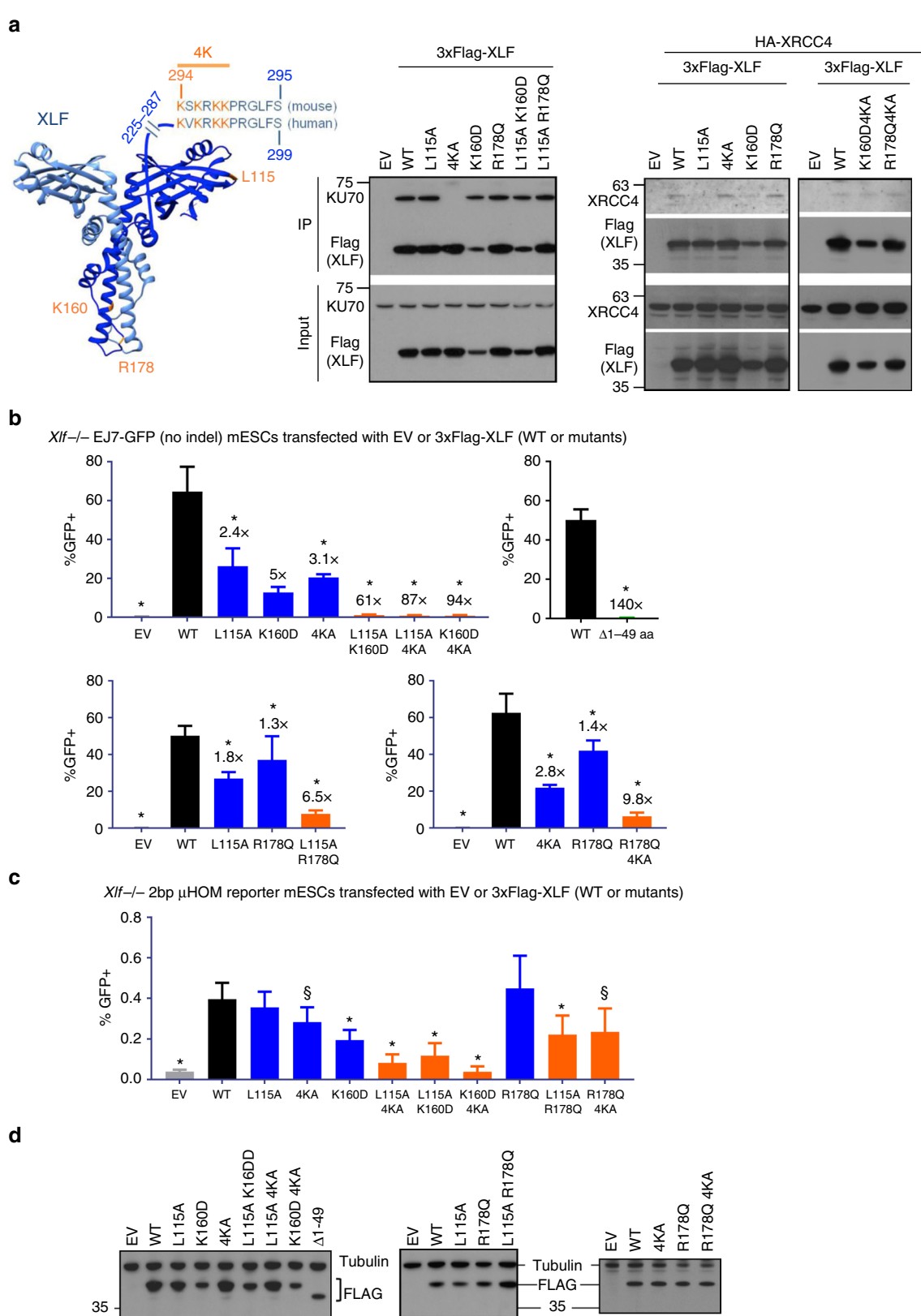

mutations (Fig. 5b). We also examined the XLF mutants on a distinct EJ event that is also promoted by XLF (2-nt µHOM assay, Fig. 3b), and found similar results (Fig. 5c). Expression of each XLF mutant was confirmed by immunoblotting (Fig. 5d).

Thus, each double mutation combination appears to cause a synergistic reduction in XLF function, compared to the single mutations. Accordingly, these results indicate that the ability for XLF to interact with either XRCC4 or KU/DNA is critical for EJ. Furthermore, disrupting one of these binding interfaces sensitizes XLF to the integrity of the coiled-coil domain. Namely, combining L115A or 4KA, with K160D or R178Q, further impairs the ability of XLF to promote EJ.

We also performed immunofluorescence analysis of 3xFlag-XLF WT and each mutant in *Xlf−/−* mESCs, by examining Flag and 4′,6-diamidino-2-phenylindole (DAPI; DNA dye) staining, particularly since XLF has been found to localize to both the nucleus and the cytoplasm[41]. We found that Flag staining was diminished in at least part of the DAPI region, which was often limited to areas of high-density DAPI staining, but in other cases reflected a more general reduction in nuclear staining (Supplementary Fig. 3a, b). These findings indicate that XLF in mESCs is present in both the nucleus and the cytoplasm, and is not distributed evenly in the nucleus, which is consistent with prior findings that this factor is regulated at the level of subcellular localization[41]. In any case, the distribution of staining patterns was not obviously different between WT XLF vs. the mutants (Supplementary Fig. 3a, b).

**Molecular dynamics simulations of XLF.** Finally, we performed all-atom molecular dynamics (MD) simulations of single and double mutants of L115A, R178Q, and K160D, alongside WT. The purpose of these simulations is to provide atomic-level insights into the dynamics of the XLF dimer structure in the presence of these mutations. The all-atom MD simulations were performed starting from the crystal structure of human XLF from residues 1 to 230, which lack the C-terminal domain[20] (Supplementary Table 1). Specifically, we focused on two aspects of the dimer interface: the distance between the coiled-coil centered around the region of K160; and a pi-pi stacking interaction between the Y167 residues on opposing monomers (Y167-Y167′) (Fig. 6a, b). The residues in the second monomer of the XLF dimer are indicated with a prime symbol. These inter-residue distances were chosen for detailed analysis, since we observed structural changes in these regions while visualizing the MD trajectories and these residues are within direct interaction distances from the mutated residues (K160D and R178Q). We found that the K160D mutant showed an increase in both the coiled-coil distance, and the Y167-Y167′ distance, compared to WT (Fig. 6a, b, Supplementary Table 1). Along with this increase in these distances, our simulations also found an infiltration of water molecules into the coiled-coil region at K160D (Supplementary

Table 1). For R178Q, we observed a slight increase in the coiled-coil and the Y167-Y167′ distances, whereas L115A was not distinct from WT. These findings indicate that K160D, and to a lesser extent R178Q, affects the dynamics of the dimer interface.

Interestingly, L115A/K160D showed both a greater coiled-coil distance and a slight increase in the Y167-Y167′ distance compared to WT, but neither was elevated to the same extent as K160D. Furthermore, L115A/R178Q was not distinct from WT in either measurements. These findings indicate that the region surrounding L115 could have long-range effects on XLF dynamics that could extend into the coiled-coil region. To investigate this further, we calculated the strength of the allosteric communication pipelines between residues in the vicinity of L115 in the globular head domain and residues in the vicinity of K160 and R178 within the coiled-coil region (Supplementary Fig. 4). We found strong pipelines of allosteric communication connecting these regions in both monomers, in the WT and mutant dimers. Interestingly, we observe higher strength of these pipelines in all of the mutant dimers than we do in the WT (22–83% increase). These findings reinforce the notion that in addition to affecting individual interaction interfaces, mutations can also have effects on the overall dynamics of the protein, including long-range effects.

## Discussion

We have sought to examine the fidelity of C-NHEJ, finding that this pathway is essential for EJ between distal ends of two chromosomal DSBs without causing indels (Fig. 7). Our findings are consistent with studies examining double-stranded oligonucleotides that are electroporated into mammalian cells, which can be rapidly ligated without causing indels, and are substantially dependent on LIG4[43]. Similarly, distal EJ between tandem I-SceI DSBs that uses the complementary 3′ overhangs to restore the I-SceI site requires KU[7,44]. Thus, for DSB ends that can be ligated without end processing, C-NHEJ appears to be essential for joining such ends without causing indels. A corollary of these findings is that C-NHEJ does not appear to be intrinsically prone to causing indels.

Accordingly, the probability that C-NHEJ will cause a mutation is likely determined by the nature of the DNA lesion, such as DSBs that are associated with loss of nucleotides, base damage, or protein crosslinks[2,45,46]. Processing such damage by nucleases/polymerases to form ends that are readily repaired by C-NHEJ is a likely source of indels. In addition, for endonuclease-generated single DSBs, the frequency of indels may be affected by the relative persistence of these lesions, in that restorative EJ of a single DSB recreates the cleavage site, which is prone to repeated cycles of DSB formation and repair[47,48]. In contrast, such cycles of breakage/repair are probably less frequent in our reporter assay, because distal EJ does not restore an sgRNA/Cas9 cleavage site.

**Fig. 5** Mutations in distinct domains of XLF have a synergistic effect on distal EJ without indels. **a** Shown is the known structure of the XLF dimer (aa 1–224, using information from the Protein Data Bank, Code 2R9A, image generated with UCSF Chimera), with the two monomers in dark and light blue. The unstructured C terminus (aa 225–299) is drawn using a blue line. Residues that are being examined in this study are highlighted in orange. Also shown are effects of various XLF mutations on complex formation with KU70 and XRCC4. Lysates were prepared from *Xlf−/−* mESCs transfected with WT or mutant 3xFlag-tagged XLF (mouse) expression vectors. Experiments examining XRCC4 included an HA-XRCC4 expression vector. A fraction of the lysate was used for the input, and the rest was used for a Flag-immunoprecipitation (Flag-IP). Shown are immunoblot signals for Flag (XLF), KU70, and XRCC4. Numbers denote molecular weight marker positions (kDa). **b** Shown is the frequency of GFP+ cells of EJ7-GFP in *Xlf−/−* mESCs transfected with control EV, or 3xFlag-XLF expression vectors for WT and various mutants. $N = 6$, error bars indicate s.d. Also shown are the fold changes relative to complementation with WT. *$p < 0.05$, mutants compared to the WT using unpaired *t*-tests with the Holm-Sidak correction, except WT vs. Δ1–49 aa was compared using an unpaired, two-tailed *t*-test. **c** Shown is the frequency of GFP+ cells for the 2-nt microhomology reporter in *Xlf−/−* mESCs transfected with control (EV), or 3xFlag-tagged expression vectors for WT or mutant XLF. *$p < 0.04$, mutants compared to the WT using unpaired *t*-tests with the Holm-Sidak correction. §$p < 0.05$, but >0.05 when adjusted for multiple comparisons. **d** Shown are immunoblots examining expression of 3xFlag-XLF WT and the various mutants

For EJ events leading to indels, either C-NHEJ or Alt-EJ could mediate such repair. Defining the pathway that causes such mutagenic events, particularly those associated with chromosomal rearrangements, is important for understanding cancer etiology. However, ascribing particular indel events to either pathway is challenging, although microhomology usage can provide some insight. Thus, we sought to examine the demarcation for microhomology requirements between these pathways. We found that C-NHEJ is critical for EJ using 1–2 nts of terminal microhomology, but is relatively dispensable for events involving 3–4 nts (Fig. 7). Furthermore, we found that KU70 substantially inhibits EJ using 4 nts of microhomology that is embedded from the edge of the DSB. In contrast, we found that two mediators of the Alt-EJ pathway (POLQ and CtIP) are important for such EJ. These findings are consistent with other reports that POLQ and CtIP are important for EJ events with deletion mutations that use microhomology[7,49,50]. We also found that these factors are dispensable for EJ without indel mutations or that use 1–2 nts of

terminal microhomology. Thus, POLQ and CtIP have the opposite effect of C-NHEJ factors on several EJ events. Notably, for EJ events involving 3–4 nts of terminal microhomology, we find that neither POLQ and CtIP nor C-NHEJ factors are absolutely required for these events. Thus, while C-NHEJ appears dispensable for EJ using 3–4 nts of terminal microhomology, there is likely redundancy between C-NHEJ and Alt-EJ for such repair. Consistent with this notion, several reports have identified redundancy between these pathways (specifically, POLQ and C-NHEJ) for some EJ events, such as random integration of exogenous DNA[37,40,51,52]. In any case, our findings indicate that chromosomal rearrangements with 0–2 nts of microhomology are more likely generated by C-NHEJ than Alt-EJ.

As part of our examination of C-NHEJ factors, we found that XLF is important for distal EJ without indels, in both mESCs and a human fibroblast cell line. This finding is consistent with the activity of XLF to promote XRCC4-LIG4 activity in vitro, as well as the substantial radiosensitivity of XLF-deficient cells[11,17,28].

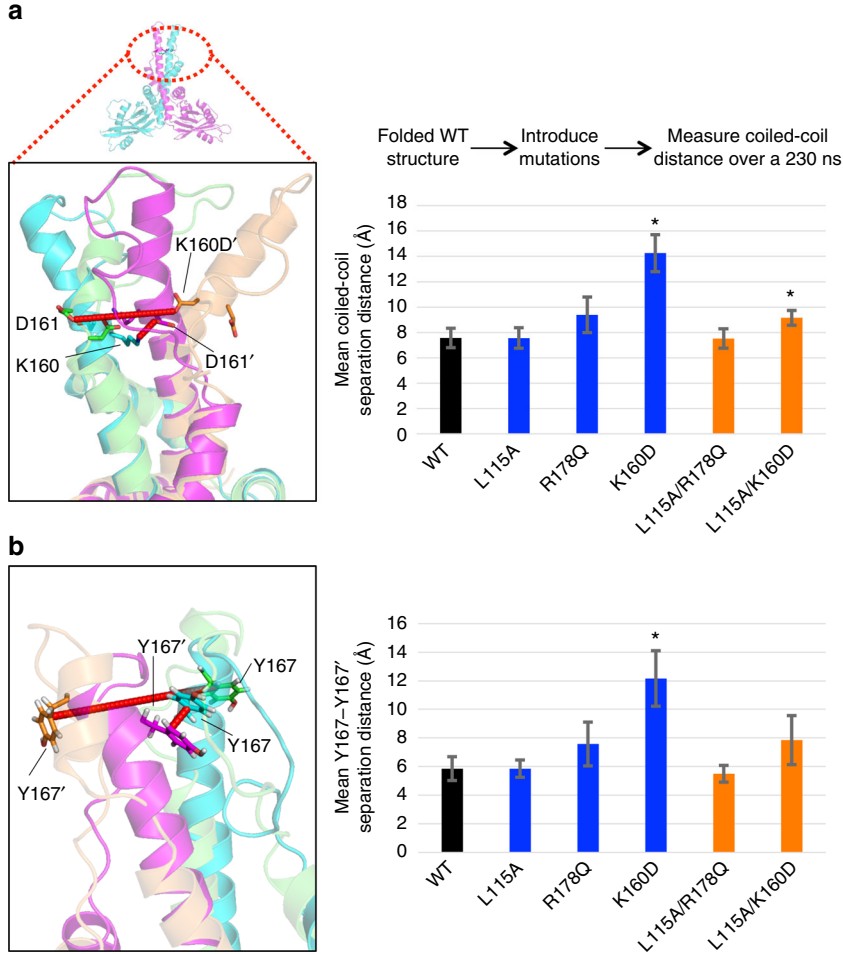

**Fig. 6** Molecular dynamics simulations of the dimer interface for XLF WT and several mutants. **a** We measured the distance between the coiled-coil domains of each monomer of the XLF dimer for WT and mutant dimers from individual replicates of 230 ns simulations. A representative figure (left) shows the extreme distances of the separation between the coiled-coil domain observed in simulations: the WT structure (cyan and magenta dimer pair) in which the dimer interface is kept intact, and the K160D mutant (green and orange dimer pair) in which the dimer interface is disrupted. We observe an increase of ~6 Å in the distance between the coiled-coil interface of each monomer in the K160D mutant compared to the wild type. Also shown (right) are the distances for the coiled-coil domain, centered near residue K160, as the mean of replicates. Error bars show the 95% confidence interval from a one-directional Student's *t*-test distribution. Asterisks denote significant difference compared to the WT. **b** Shown (left) are representative extreme distances for the pi stacking interaction of Y167 and Y167' in the WT (cyan and magenta dimer pair) and the R178Q mutant dimer (green and orange dimer pair). Also shown (right) is the distance between the center of mass of Y167 and Y167', plotted as mean of replicates. Error bars represent the 95% confidence interval from a one-directional Student's *t*-test distribution. Asterisks denote significant difference compared to the WT

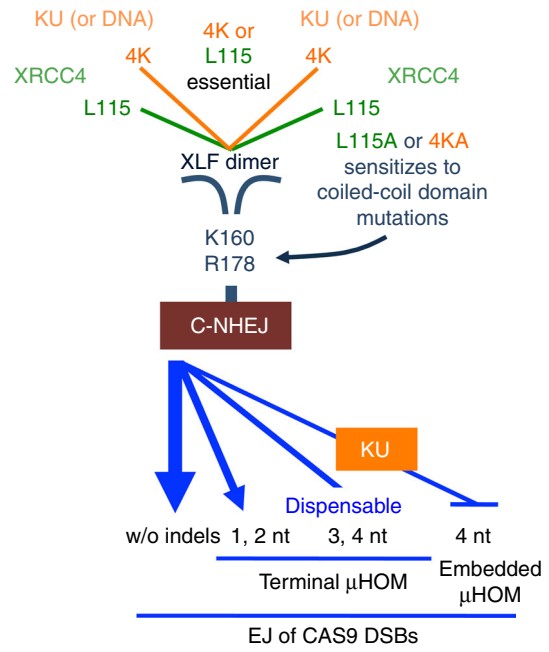

**Fig. 7** Summary. Shown is a model for the function of distinct XLF domains during C-NHEJ, as well as the role of C-NHEJ in distinct EJ events

However, in vitro experiments have found that XLF is dispensable for XRCC4-LIG4 joining of blunt ends, and instead may be particularly important for non-complementary ends[15,30]. Since these studies appear to contradict our findings, we suggest that protection and bridging of DSB ends to avoid indels may have a more stringent requirement for XLF in the context of the cell, compared to in vitro ligation assays. Genetic studies have also supported the notion that XLF has a more specialized role during C-NHEJ. Namely, XLF appears to show substantial redundancy with several factors for V(D)J recombination, including the KU-binding protein PAXX[10–16,31–35]. In contrast, while we found that PAXX promotes distal EJ without indels, its influence on this event was two orders of magnitude lower than that for XLF. Furthermore, we found that PAXX expression was unable to rescue this EJ event in $Xlf-/-$ cells. To reconcile our findings with reports that XLF and PAXX have overlapping functions during C-NHEJ, we speculate that such redundancy may be dependent on the context of the C-NHEJ event, such as repair requiring DSB end processing prior to ligation.

Along these lines, it is also conceivable that XLF has a unique requirement for EJ without indels between two DSBs (i.e., distal EJ). However, a prior study from our group demonstrated that XLF is dispensable in mESCs for distal EJ per se, based on a reporter for a 0.4 megabase pair deletion rearrangement induced by two DSBs[22]. Furthermore, this study showed that XLF deficiency affects indel mutation patterns at a single DSB (e.g., caused an increase in the size of deletion mutations)[22].

Regarding the precise role of XLF during C-NHEJ, a variety of mechanisms are possible. For one, XLF can form large filamentous complexes with XRCC4, which have a remarkable DNA-bridging capacity that permits sliding of the two DNA molecules (i.e., the sliding sleeve model)[53]. Alternatively, XLF could act as a bridge between two C-NHEJ complexes bound to DSB ends, in that the XLF dimer has the potential to interact with two molecules of XRCC4, KU, and/or DNA. Such a bridging function could shorten the distance between DSB ends to favor ligation, which is consistent with single-molecule studies[54]. We have found the L115A mutant, which is deficient for the interaction with XRCC4[17], is relatively proficient at the EJ event

measured in our study. This result is consistent with other functional analysis of this mutant[12,17,21]. These findings appear inconsistent with a primary role for XLF as forming large filamentous complexes with XRCC4. In contrast, combining L115A with mutations in C-terminal conserved lysines, which are important for binding to KU and/or DNA[17–19], abolished XLF function to promote distal EJ without causing indels. We suggest that these findings support a role for the XLF dimer as a bridge between two C-NHEJ complexes (Fig. 7). Such end bridging could be particularly important to join DNA ends that are not stabilized by an annealing intermediate, and hence is consistent with having a critical role for EJ of blunt DSBs without causing indels.

Consistent with this model, we find that the L115A and 4KA mutants are particularly sensitive to mutations at the dimer interface: K160D and R178Q. The K160D mutant affects the dimer interface, since it disrupts the salt bridge between K160 and D161′ on opposing monomers. The cancer-associated[41,42] R178Q mutation could also potentially affect the dimer interface, since this residue is within the coiled-coil region. Indeed, our MD simulations found that the K160D mutation causes a significant separation of the distance between the monomers at the coiled-coil interface, and at a pi-pi stacking interaction at Y167-Y167′. The R178Q mutation also affected these distances, but to a lesser extent. In summary, we speculate that disrupting the interaction of XLF with either XRCC4 or KU/DNA leaves only one bridging interface, which if combined with disruption of the dimer interface, could severely weaken its bridging capacity.

Finally, we also compared the effects of single and double mutants on XLF structural perturbations using MD simulations, to capture the early events that lead to structural perturbations caused by the mutations. We found that the L115A/K160D mutant also showed an increase in mean coiled-coil and Y167-Y167′ distances compared to WT, but to a lesser degree than K160D. It is unlikely that the L115A mutation rescues the dimer interface repulsion that is caused by K160D. Instead, the ensemble of conformations initiated by the K160D single mutation could be more dynamic than in the context of the double mutant, due to the increase in correlated movement between multiple mutations. Consistent with this notion, we observed stronger long-range allosteric communication between residues around L115A and the residues in the coiled-coil domain. Indeed, the strength of the allosteric pipelines increased in each of the single and double mutant dimers as compared to the WT. While the implications of these allosteric interactions on XLF function during C-NHEJ remain unclear, these findings indicate that in addition to affecting particular binding interfaces, individual mutations can also affect the long-range dynamics of XLF structure, including across different domains.

In conclusion, since distal EJ without indels is substantially dependent on several core C-NHEJ factors, including XLF, we suggest that further characterizing the mechanism of such EJ, using the EJ7-GFP reporter assay system, will provide insight into genome stability.

## Methods

**Plasmids, cell lines, and siRNA**. The pCAGGS-EJ7-GFP reporter, pCAGGS-3xFlag-hXLF, and pCAGGS-3xFlag-PAXX vectors were generated by cloning gBLOCK fragments (IDT) into the pCAGGS-BSKX expression vector[22]. The pCAGGS-EJ7-GFP cassette was then introduced into the Pim1-targeting vector containing a promoterless hyg gene[22]. The variants of EJ7-GFP with micro-homology were introduced as gBLOCKs (IDT) into this pim-EJ7-GFP vector. These plasmids were used to target the reporters to the Pim1 locus using electroporation of linearized plasmids, selecting cells with hygromycin, and screening for targeted clones by PCR with the following primers (Pim1Ex1F: 5′-AAGAT CAACTCCCTGGCCCACCTGCG-3′; Pim1Ex4R: 5′ TGTTCTCGTCCTTGA TGTCG-3′; Hyg3A: 5′-CCGCTCGTCTGGCTAAGAT-3′)[22]. EJ7-GFP was also introduced to pCDNA5/FRT (Thermo Fisher), which was integrated into the FRT

locus of U2OS Flp-In T-REx cells, using co-transfection with the PGK-FLP vector[55], and selection of individual clones in 0.2 mg ml⁻¹ hygromycin. The pCAGGS-3xFlag-XLF mouse expression vector was generated by introducing a gBLOCK fragment with the 3xFlag sequence into pCAGGS-XLF[22]. Mutant forms of this expression vector were generated by inserting gBLOCK fragments or double-stranded oligonucleotides. The pCAGGS-hLIG4 plasmid was generated using the pDR125 plasmid (Addgene 37150, generously deposited by Dr. Dale Ramsden). All sgRNAs were expressed from the px330 plasmid, which also co-expresses Cas9 (Addgene 42230, generously deposited by Dr. Feng Zhang)[56]. We used a non-targeting siRNA (siControl, GE Dharmacon D-001810-01), and pool of four siRNAs targeting CtIP (siCtIP, GE Dharmacon, D-055713-14, -15, -16, and -17).

Several of the mESC cell lines, along with culture conditions, were described previously (WT, *Xlf*−/−, *Xrcc4*−/−, and *Ku70*−/−)[10,25,26]. The 2BN XLF-deficient cell line[28], the U2OS Flp-In T-REx[27], and the *Polq*−/− mESC line with the Flag-POLQ vector[40] were generously provided by Dr. Penny Jeggo (University of Sussex), Dr. Tanya Paull (University of Texas), and Dr. Roland Kanaar (Erasmus University Medical Center), respectively. The U2OS line was validated by short tandem repeat profiling. The *Polq*−/− genotype was confirmed with PCR (5′-TTGGGGACTTCCTAAAGCAG-3′ and 5′-CCAACCCAAAAGAGAATT GTTC-3′). The 411BR (*LIG4* hypomorph cell line)[29] and control fibroblast lines were acquired from the Coriell Biorepository (GM16089 and GM00637, respectively). The *Paxx*−/− mESC line was generated by co-expressing two previously described sgRNAs with Cas9 to excise the gene (sgRNAs: 5′-CTAAGGT GTTCGCTCGGCGG-3′ and 5′-CAGTTTATTTGACGGAGAA-3′)[16]. Clones were also transfected with a dsRED expression plasmid, dsRED+ cells were sorted, plated at low density, and clones screened by PCR for deletion of the gene (5′-ATT GAAGAGCGGCAGATATGT-3′ and 5′-ACGCAGAATCAACACAGTAGGT-3′). Loss of PAXX mRNA expression was confirmed with RT-PCR with these primers (5′-ATGGCTCCTCCGTTGTTGTC-3′ and 5′-GCGTCCGTCACATAGAGGTT-3′) and ACTIN control (5′-GGCTGTATTCCCCTCCATCG-3′ and 5′-TCTCCA GGGAGGAAGAGGAT-3′). Cell lines tested negative for mycoplasma contamination using the Lonza Mycoplasma Detection kit.

**DSB repair assays.** For the DSB repair assays, mESCs, U2OS, or human fibroblast cell lines were seeded at a cell density of $0.5 \times 10^5$ cells per well of a 24-well plate. Each well was transfected with 200 ng of each sgRNA/Cas9 plasmid and 50 ng of the control EV or complementation vector using 3.6 µl of Lipofectamine 2000 in 0.5 ml of antibiotic-free media. Extrachromosomal plasmid assays included 200 ng of the reporter plasmid (in the *Pim1*-targeting vector). For siCtIP experiments, cells were seeded on a mixture of 3.75 pmol of siRNA with 1.8 µl of RNAiMAX (Thermofisher), and 3.75 pmol of siRNA was included in the transfection with the sgRNA/Cas9 plasmids. For immunoblotting analysis, the siRNA experiment was scaled fourfold, and the pgk-puro plasmid (600 ng) and EV (1200 ng) replaced the sgRNA/Cas9 plasmids, and after 1 day cells were selected in puromycin (1.8 µg ml⁻¹, Sigma) for 2 days.

Cells were analyzed by flow cytometry 3 days post transfection. The frequency of GFP+ cells was normalized to transfection efficiency, which was measured using parallel transfections with a GFP-expression vector (pCAGGS-NZE-GFP)[57]. For example, if a Cas9/sgRNA transfection resulted in 3% GFP+ cells, and the transfection frequency for that cell line in the parallel assays is 30% GFP+, then the normalized repair frequency is 10% GFP+. Each bar represents the mean of at least five independent transfections, error bars represent standard deviation, and statistics are as described in the figure legends. No data/experiments were excluded if control transfections were valid (e.g., transfection efficiency). As described in the figure legends, statistics were performed using unpaired, two-tailed *t*-tests, with corrections for multiple comparisons by the Holm-Sidak method as appropriate. Isolation of GFP+ cells were performed on either an Aria III or Aria SORP instrument (Becton Dickinson), and the product analyzed by PCR (5′-TTCCTACAGCTCCTGGGCAACG-3′ and 5′-AAGTCGTGCTGCTTCATGTG-3′).

**Immunoprecipitation and immunoblotting.** For immunoblotting analysis, cells were lysed with NETN buffer (20 mM TRIS (pH 8.0), 100 mM NaCl, 1 mM EDTA, 0.5% IGEPAL, 1.25 mM dithiothreitol and Roche Protease Inhibitor), and followed by several freeze/thaw cycles. Blots were probed with the following antibodies at a dilution of 1:1000 unless otherwise noted: FLAG (Sigma A8592); XRCC4 (1:500, Santa Cruz sc-8285); XLF (Bethyl A300-730A); KU70 (Santa Cruz sc-1487 or Cell Signaling D10A7); LIG4 (Santa Cruz sc-271299); PAXX (Abcam ab126353); CtIP (Active Motif 61141); ACTIN (1:3000, Sigma A2066); TUBULIN (Sigma T9026); and horseradish peroxidase (HRP)-conjugated secondary antibodies (1:3000, Abcam ab205718, ab205719, or ab205723). ECL reagent (Amersham Biosciences) was used to develop HRP signals. For co-immunoprecipitation analysis to examine associations of 3xFlag-XLF with KU70, $10^6$ cells were seeded onto a 10 cm plate, and then transfected with 3 µg of control (EV), 3xFlag-XLF (WT or mutant) expression vector using 12 µl of Lipofectamine 2000 in 5 ml of media without antibiotics. To examine associations of 3xFlag-XLF with XRCC4, 600 ng of pCAGGS-HA-XRCC4 was included in the transfection. Two days post transfection, cells were lysed in NETN buffer supplemented with phosphatase inhibitor (Roche PhosSTOP). Cell lysates were dounce homogenized and incubated with 25 µl of anti-FLAG M2 magnetic beads (Sigma M8823) overnight at 4 °C. Beads were washed twice with the NETN buffer and eluted with 100 mM glycine (pH 2.5), and

neutralized with 1 M Tris-HCl (pH 10.8). Blots were probed with the antibodies and developing reagents described above. Images of full blots have been provided (Supplementary Figs. 5, 6, 7).

**Immunofluorescence.** Cells were transfected as for the reporter assays, but with the 3xFlag-XLF expression vectors only, incubated for 2 days, and detached with trypsin that was then neutralized with media. Cells were affixed to slides using a Cytospin 4 (Thermofisher), fixed with 4% paraformaldehyde, permeabilized, and stained with the antibody against the Flag epitope (1:500 dilution, Sigma F3165), AlexaFluor 488 goat anti-mouse secondary antibody (1:250 dilution, Invitrogen A11029), and DAPI using Vectashield Mounting Medium (Vector Laboratories H1500). Images were acquired using the Zeiss LSM 700 Confocal Microscope at ×20, with the ZEN Black image acquisition software.

**Computational modeling and MD simulation.** All-atom MD simulations of the XLF dimer was performed starting from the 2.3 Å crystal structure (PDB ID: 2QM4)[20] of the truncated (amino acids 1–230) WT dimer. Seleno-methionine residues were mutated to methionine using Maestro Protein Preparation Wizard[58] and missing residues in monomer B (residues 170–173 and 228–230) were added using homology modeling method, MODELLER[59] using the corresponding regions in monomer A as template. Mutant XLF dimers were generated by mutating residue positions in Maestro. The dimers were then subjected to energy minimization using Maestro MacroModel (Schrödinger Release 2017-4: MacroModel, Schrödinger, LLC, New York, NY, 2017). XLF dimers were placed at the center of a simulation box, with a minimum distance between protein-edge and box-edge of 10 Å and solvated with bulk explicit TIP3P water[60] (average of 25 800 water molecules per system) and 22–26 sodium ions to neutralize the system charge, using tleap[61]. SHAKE algorithm was used to constrain bonds involving hydrogen atoms, and default parameters for PME were used to calculate long-range Coulomb interactions. System minimization, equilibration, and production dynamics were performed using the GPU version of PMEMD module of Amber14 with the ff14sb force field[61]. Each system was minimized for 15 000 steps using a non-bond cutoff of 12 Å. The first 5000 steps were performed with 5 kcal mol⁻¹ Å⁻² positional restraints on all protein atoms (water molecules unrestrained), followed by 5000 steps with 5 kcal mol⁻¹ Å⁻² positional restraints on protein backbone atoms, and finally 5000 steps with no positional restraints. The system was equilibrated for 100 ps at 1 fs time step with 5 kcal mol⁻¹ Å⁻² positional restraints on protein backbone atoms, to a temperature of 310 K under NVT ensemble using Langevin thermostat. The system was heated from 0 to 300 K in steps of 60 K with 20 ps of simulation at each step. Further equilibration was performed with an NPT ensemble at 1 bar using the Berendsen barostat while applying protein backbone restraints. The backbone restraints were started with a force constant of 5 kcal mol⁻¹ Å⁻² and reduced to 1 kcal mol⁻¹ Å⁻² in steps of 1 kcal mol⁻¹ Å⁻² for every 10 ns. This was followed by 10 ns NPT simulation with no position restraints. Production MD was initiated as four individual trajectories for each system beginning from the last frame of the unrestrained 10 ns NPT ensemble, each initiated with a new, random velocity. Each replicate was carried out to 250 ns for a total of 1 µs of simulation time. We performed 1 µs of MD simulations each on the WT XLF dimer, single mutants K160D, R178Q, and L115A, and the double mutants K160D/L115A and L115A/R178Q.

**Analysis of MD simulation trajectories.** The trajectories from the MD simulations were analyzed using the cpptraj from the AmberTools 15 package[61]. In order to focus analysis on an equilibrated ensemble of conformations for each dimeric system, we took the last 230 ns of the 250 ns simulation trajectory for each replicate. This produced a total ensemble of 920 ns per system. The plots of the root mean square deviation in coordinates of the backbone atoms from the equilibrated structure with time, for various systems, are shown in Supplementary Fig. 8. To assess changes in the stability of the coiled-coil dimerization interface, we calculated the distance between the coiled-coil region of the two XLF monomers. This distance was calculated using the center of mass of the backbone atoms making up the coiled-coil region in the vicinity of the residue K160. Thus, we calculated the distance between the center of mass of the two monomers in the region between residues 157 and 171. The average distance for the coiled-coil and the Y167-Y167′ distances were calculated for each replicate and the mean was taken for four replicates. The standard error was calculated and multiplied by the *t* value of 2.35 for three degrees of freedom at 95% confidence level. Water density was calculated using a 3 Å distance cutoff around the specific residues in the coiled-coil interface using a custom tcl script and vmd[62]. Data histograms were plotted using Matplotlib software[63]. Allosteric communication analysis was performed with *Allosteer*[64,65], an in-house developed software package, which calculates mutual information in the dihedral angle movements between amino-acid residues across the dimer. Briefly, we first calculated mutual information in dihedral angle movement[64,66], between the set of residues near the L115 and the set of residues in the vicinity of residues K160 and R178. We then used the graph theory algorithm *Allosteer*, developed by us[65], to calculate the allosteric communication pipelines between these sets of residues. Pymol (The PyMOL Molecular Graphics System, Version 1.6 Schrödinger, LLC.) was used to visualize the results of allosteric communication analysis, and to generate representative images from MD simulation. The strengths

of the pipelines were compared between WT and each mutant to determine changes in the allosteric communication between WT and mutants. The significance of these differences was measured using a Wilcoxon rank-sum test. We have also measured the dynamic cross-correlation of amino-acid residues in the WT and mutant XLF dimers to observe the dynamics of positively correlated amino-acid pairs in each dimer (Supplementary Fig. 9). This analysis was performed using cpptraj in AmberTools 15 package and we determined no significant differences between WT and mutant simulations through this analysis.

**Data availability**. The data sets generated during and/or analyzed during the current study are available from the corresponding author on reasonable request. All code, including versions used, is described in the above sections, and is either commercially available or published.

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

## Acknowledgements

This study was funded in part by the National Cancer Institute and National Institute of General Medical Sciences of the National Institutes of Health: R01CA120954 and R01CA197506 (J.M.S.); R01GM097261 (N.V.); and P30CA33572 (for City of Hope Core Facilities).

## Author contributions

R.B., M.S., N.V., and J.M.S. designed research; R.B., S.M., G.L., and J.M.S. performed research; all authors analyzed data; R.B. and J.M.S. wrote the paper with input from all authors.

## Additional information

**Competing interests:** The authors declare no competing interests.

