## [Peer Review File · Nature Communications]

Reviewers' Comments:

Reviewer #1:

Remarks to the Author:

Bhargava et al analyzed classical non-homologous end joining events (C-NHEJ) in mammalian (mouse and human) cells using an assay that entails inducing two blunt ended Cas9 DNA breaks at DNA sequence that interrupts GFP gene. End joining without any insertions and deletions (indels) at the junctions restores GFP coding sequence and render cells green fluorescent. Such end joining depends on Ku, XLF, Lig4, and XRCC4, key C-NHEJ factors. The authors then modified the construct to examine alternative-end joining (A-EJ) events that use 1-4 bp microhomology (MH) for annealing. The end joining events mediated by annealing of flanking 4 bp MHs do not depend on Ku or XLF, raising the possibility that the system allows the investigators to discriminate between these two types of end joining events genetically. The assay is elegant and it should be useful to decipher distinct genetic requirements if the events mediated by 4 bp MHs are indeed A-EJ.

The reporter system also offers a unique opportunity to elucidate the role of XLF in C-NHEJ. Previous biochemical study showed that XLF stimulates XRCC4-Lig4 activity but is dispensable for C-NHEJ. Part of this has been attributed to functional redundancy between XLF and PAXX. Exploiting the reporter that depends on XLF for repair, the authors showed that dimerization of XLF and the interaction with XRCC4 as well as Ku are likely required for XLF function in C-NHEJ. The authors propose that XLF likely bridges DNA ends for joint formation.

1. One biggest issue is that the authors did not regard inevitable variations in targeting efficiency between different CAS9 constructs in their calculation of end joining frequency measurement. Since the experiments used multiple combination of Cas9 cleavage sites using different sets of guide RNAs, one cannot assume the targeting and cleavage efficiency of CAS9 are identical and therefore the GFP+ frequency could reflect targeting and cleavage efficiency variations among these set-ups. For instance, the authors came to the conclusions that A-EJ frequency is lower than that of C-NHEJ because the former produces less GFP+ cells. Furthermore, the frequency of A-EJ is elevated in extrachromosomal reporter construct. Part of this outcomes could be readily explained by the variations in cleavage efficiency. To resolve this issue, the author should consider testing and validating the frequency of end joining in each gRNA-CAS9 combination in simpler end joining assay (a single CAS9 induced break) and compare their values to be equivalent or similar to each other.
2. The types of events mediated by 4 bp MH do not depend on Ku and XLF. Are they dependent on CtIP or Lig3? Additional genetic testing is needed to verify if the described event is indeed A-EJ.
3. Overexpression of PAXX did not offset C-NHEJ defect in XLF-/- cells. However, since the authors cannot monitor the level of PAXX by Western blot, one cannot tell if it is over-expression or even under-expression. It is still possible that the level of expression of PAXX is too low to complement XLF defect. The author should disclose the limitation of this approach more thoroughly.
4. Is L115A 4KD double mutant defective in interaction with both Ku and XRCC4? Such mutant appears to retain some end joining function in the assay (Fig. 4B).
5. Does the current system offer a unique condition that depends on XLF for C-NHEJ because it induces two tandemly placed DNA breaks? Is end joining of a single CAS9-break still dependent on XLF?
6. The authors used both integrated or extrachromosomal GFP reporters to assess the frequency of C-NHEJ and their genetic requirements. In several places, however, the authors should not compare the outcomes from these two different conditions. It is recommended to modify the manuscript to explicitly describe these conditions in details in the text so the conclusions could be

deduced only using the comparable assays.

Reviewer #2:

Remarks to the Author:

In their manuscript, Bhargave et al. study the function of Classical Non-Homologous End Joining (C-NHEJ) factors using Cas9-based end joining (EJ) assays. The authors demonstrate that XLF, XRCC4 and KU70 promote precise EJ without indel formation. They also demonstrate the requirement of the above factors for EJ events with less than 3 nts of terminal microhomology. Furthermore, they characterize XLF-mediated precise EJ activity and demonstrate the synergistic function of distinct XLF domains (XRCC4-binding domain, KU-binding domain and coil-coil domain) in repairing DNA double-strand breaks (DSBs) by EJ. These findings are of potential interest for the DNA repair and gene editing communities. The experiments are well conducted and the data are clearly presented. We invite the authors to address the minor points indicated below.

Comments:

- 1) Figure 3B. The authors convincingly demonstrate the necessary function of KU70 and XLF in EJ that utilizes 1-2 bps of microhomology and no effect of KU70 and XLF deficiency on EJ that utilizes 3-4 bps microhomology. They conclude that C-NHEJ involves EJ with less than 3 bp microhomology and Alt-EJ promotes EJ with ≥ 3 bp microhomology (discussion, page 17). To strengthen these findings, it would be important to determine whether deletion of Alt-EJ factors causes a reduction in EJ with ≥ 3 bp microhomology using the EJ assays described in Figure 3A.
- 2) Figure 4B. What is the effect of the L115A, K160D, R178Q and 4KA XLF mutants (alone and in combination) in the microhomology-mediated EJ assays described in Figure 3A?
- 3) Figures 2B and 4A. The authors should provide images with longer exposures of the XLF (Figure 2B) and XRCC4 (Figure 4A, IP) blots.

Reviewer #3:

Remarks to the Author:

This manuscript presents the development of an assay to detect end-joining to assess fidelity of canonical non-homologous EJ (C-NHEJ). The authors show that this process requires several C-NHEJ factors including XLF. The further results indicate that disruption of specific residues on one of two domains affects EJ without indels. In particular, single mutations L115A, K160D or 4KA showed reduction and double mutants showed almost complete loss of EJ function. MD simulations on WT and several single and double mutants suggest that K160D results in increase distance of the dimer interface, similar to R178Q, although to a lesser extent. The double mutation of L115A/K160D and L115A/R178Q suggest a change in allosteric communication with the former exhibiting a significant gain in strength in allosteric communication compared to the WT system. This manuscript reports a new technique that enables the investigation of a very important DNA repair process, with thorough validation and investigation of possible effects.

I will comment on the MD simulations, as this is my field of expertise. Overall, the simulations provide interesting insights on the possible effects of the point mutations. Nevertheless, the manuscript needs to include a more thorough discussion of the MD results on all the systems and connection to the experimental results, and better description of the methods. In particular:

-The authors only discuss selected results from the MD simulations for the various mutants. The manuscript would greatly benefit from increasing the scope of analysis for the simulations including:

a) distance analysis between K160D/D'161, Y167/Y'167 for all systems, i.e. what is the effect of the double mutants on the dimer interface? not only allosteric analysis, also include allosteric analysis on the single mutants.

b) The authors are encouraged to add residue-wise cross-correlation analysis to complement the analysis of the MD simulations.

c) provide statistical analysis on all the results, in the present version it is only inferred from Fig. S3a for two of the systems studied.

d) include all analysis in a condensed figure (distance, cross-correlation, statistical analysis, etc.) either in the main text or SI. Also include at least a plot of backbone RMSD to show that/if systems are equilibrated and include a mention in the main text in this regard.

e) are the combination of distance/allosteric pipeline/cross-correlation results consistent with the experimental results, specifically with respect to the observed effects of single vs. double mutants? i.e., the allosteric analysis for L115A/K160D suggest (not indicate) a possible reason for the observed effect of the double mutant. However, the allosteric analysis by itself on the double mutants and distance analysis for the single mutants as they are reported and discussed in the manuscript do not provide insights on the experimentally observed reduction in activity for the single mutants compared with the (almost) elimination of activity in the double mutant.

-The authors are encouraged to use different notation to identify residues from different monomers (see above) to improve clarity. For example: use unprimed to denote residues from one monomer and primed for another, e.g. Y176 for monomer 1 and Y176' for monomer 2.

-There are several details in the methods section that need to be included:

a) number of water molecules (or size of simulation box),

b) minimum distance from edge of protein to edge of solvent box

c) is the non-bonded cutoff the same for real-space for Coulomb and VdW? did the authors use the default parameters for PME for the long-range Coulomb interactions?

d) what thermostat/barostat were employed?

e) did the authors use the CPU or GPGPU version of pmemd?

Point-by-point responses to the reviewers' comments for the revised manuscript of Bhargava et al., NCOMMS-17-32643-T. We thank the reviewers for their suggestions to improve the manuscript. We have responded to each concern with new experiments or edits to the text, as appropriate. The reviewer comment is shown in *italics/green*, and is followed by our response.

Reviewer #1:

1. One biggest issue is that the authors did not regard inevitable variations in targeting efficiency between different CAS9 constructs in their calculation of end joining frequency measurement. Since the experiments used multiple combination of Cas9 cleavage sites using different sets of guide RNAs, one cannot assume the targeting and cleavage efficiency of CAS9 are identical and therefore the GFP+ frequency could reflect targeting and cleavage efficiency variations among these set-ups. For instance, the authors came to the conclusions that A-EJ frequency is lower than that of C-NHEJ because the former produces less GFP+ cells. Furthermore, the frequency of A-EJ is elevated in extrachromosomal reporter construct. Part of this outcomes could be readily explained by the variations in cleavage efficiency. To resolve this issue, the author should consider testing and validating the frequency of end joining in each gRNA–CAS9 combination in simpler end joining assay (a single CAS9 induced break) and compare their values to be equivalent or similar to each other.

The reviewer is concerned with our comparisons of the relative frequencies of EJ events between different reporter setups. The design of all the reporters involves two tandem Cas9-induced DSBs (using two sgRNAs) to excise a fragment between the DSBs. Distal EJ between the two DSBs that restores the GFP gene can be measured by flow cytometry. However, as the reviewer mentions, the frequency of the GFP+ product could be potentially affected by the efficiency of tandem DSB formation to excise the intervening fragment, particularly since each reporter uses a distinct set of sgRNAs.

To directly address this concern, we measured total distal EJ (i.e., excision of the fragment between the two DSBs) in each reporter / set of sgRNAs, using PCR analysis (new Supplementary Fig S1). This analysis the most direct control for the sgRNA pairs, since we are comparing total distal EJ vs. distal EJ that restores GFP+. From this analysis, we found that the no indel (EJ7) and 4 nts μ HOM reporters show similar frequencies of total distal EJ, whereas the %GFP+ frequency for the 4 nts μ HOM reporter is 12-fold lower. The 1-3 nts μ HOM reporters show lower levels of total distal EJ compared to the no indel reporter (EJ7), although the %GFP+ frequencies are reduced to a much greater extent. Thus, these findings do not affect the overall conclusion that distal EJ without indels is much more frequent than events using terminal microhomology to cause deletions. We thank the reviewer for this request to validate our reporter assays, and thereby strengthen our study.

We also wanted to provide two additional details in our response to this concern:

1. We attempted to examine mutagenesis at single sgRNA sites using the PCR/surveyor nuclease assay as suggested by the reviewer, but surveyor treatment caused background/non-specific bands that we were unable to resolve from the expected cleaved product. Thus, we found that examining distal EJ was both the most direct and feasible control for these experiments.

2. The reviewer is concerned with comparisons of plasmid vs. chromosomal assays, which is also the focus of point #6. In response to these concerns, we have removed comparisons between chromosomal and extrachromosomal plasmid assays throughout the manuscript, and instead only describe comparisons within the same condition. In the parent manuscript, we had included one comparison between these two conditions in WT cells, but as this was only a minor observation in the parent manuscript, removing this comparison does not affect the major conclusions of the manuscript.

2. The types of events mediated by 4 bp MH do not depend on Ku and XLF. Are they dependent on CtIP or Lig3? Additional genetic testing is needed to verify if the described event is indeed A-EJ.

The reviewer is interested in analysis of other factors in our new EJ reporters, particularly factors that have been implicated in Alt-EJ, and thereby provide a contrast with C-NHEJ factors. We have responded with new experiments and an expanded Discussion, as follows.

We have added new experiments that show the influence of CtIP and POLQ on our EJ reporters (new Fig 4). We chose to examine POLQ instead of LIG3, since while POLQ has been directly implicated in Alt-EJ, LIG3 appears to be functionally redundant with LIG1 for EJ (e.g., PMID 28687761 and PMID 26787901).

From this analysis, we find that CtIP and POLQ are each important for EJ using 4 nts of microhomology that is embedded from the edge of a DSB. In contrast, CtIP depletion causes a modest increase in EJ using 1-2 nts of terminal microhomology, and EJ without indels. Similarly, POLQ-deficiency causes a modest increase in EJ using 1 nt of terminal microhomology, and has no effect on EJ without indels or using 2 nts of terminal microhomology. Accordingly, CtIP and POLQ have distinct effects vs. C-NHEJ factors on these EJ events. Notably, for EJ using 3-4 nts of terminal microhomology, CtIP has no influence, and POLQ is only modestly important for these events. Accordingly, we have edited the Discussion that describes distinctions between Alt-EJ and C-NHEJ (paragraph 3). As an added note, to strengthen the analysis of C-NHEJ factors on these EJ events, we have added experiments with *Xrcc4*^{-/-} mESCs and the microhomology reporters (new Supplementary Fig S2b), and found similar results to KU70 and XLF.

3. Overexpression of PAXX did not offset C-NHEJ defect in XLF^{-/-} cells. However, since the authors cannot monitor the level of PAXX by Western blot, one cannot tell if it is over-expression or even under-expression. It is still possible that the level of expression of PAXX is too low to complement XLF defect. The author should disclose the limitation of this approach more thoroughly.

The reviewer is concerned that since we cannot detect endogenous mouse PAXX by immunoblot, we should raise the limitations of this approach in the Results section. We have responded with the following edits. For one, we clarified that our expression vector is mouse PAXX, such that the possibility that we are under-expressing PAXX is unlikely. Nevertheless, we have also directly addressed the limitation raised by the reviewer, by adding this sentence to the Results “Although, we were unable to quantify the fold-effect on PAXX levels in these experiments, since we cannot detect endogenous mouse PAXX by immunoblotting.” Finally, we have replaced the phrase “overexpression of PAXX” with “expression of PAXX.”

4. Is L115A 4KD double mutant defective in interaction with both Ku and XRCC4? Such mutant appears to retain some end joining function in the assay (Fig. 4B).

The reviewer requests a discussion of the apparent residual EJ activity of the L115A/4KA double mutant form of XLF, since the L115A mutation disrupts the interaction with XRCC4, and the 4KA mutations disrupt binding to KU. We have responded in three ways.

1. We provide a new co-IP experiment with the L115A/4KA double mutant (new Supplementary Fig S2c) to confirm the lack of a detectable complex with KU70 or XRCC4.

2. L115A/4KA mutant shows an 87-fold reduction in EJ frequency vs. WT. Since we used a broken y-axis to show the data in the parent manuscript, this substantial reduction in EJ function may not have been visually apparent. Thus, in response, we have used a continuous y-axis for all the plots in this figure (now Fig 5), and thereby provide a more clear representation of the data.

3. In addition, the reviewer is correct that expression of XLF L115A/4KA induces a frequency of GFP+ cells above background levels (2.7-fold above empty vector). The other double-mutants also show similar residual activity. These data reflect the wide dynamic range of this reporter assay, which is due both to the very low background level of GFP+ cells without XLF (i.e. the empty vector transfections), and the substantial complementation with WT XLF. Accordingly, we can distinguish between L115A/4KA and empty vector, which show an 87-fold and >200-fold reduction vs. WT, respectively. In contrast, other assays may not be sensitive enough to detect such low-level residual activity. So, in response, we have also added this clarification sentence to the Results: "Although, the double-mutants retained residual activity above background, reflecting the wide dynamic range of this assay (see Fig 1b), compared to other assays, such as co-IP."

5. Does the current system offer a unique condition that depends on XLF for C-NHEJ because it induces two tandemly placed DNA breaks? Is end joining of a single CAS9-break still dependent on XLF?

The reviewer is interested in whether XLF is important specifically for EJ without indels between two DSBs (measured by our reporter), vs. a role for such EJ at a single DSB. Examining EJ of a single DSB that fails to cause indels is technically unfeasible, since the product of such EJ is not distinct from a site that was never cleaved. Accordingly, it is not possible to develop a reporter assay for such EJ at a single DSB. Examining the EJ events using 1-2 nts of terminal microhomology also requires the setup of two DSBs, due to the restrictions on the sequence context of CAS9-induced DSBs vs. the GFP coding sequence (e.g., the position of the protospacer adjacent motif sequence).

Thus, we have responded with two additions to the text. For one, we have clarified the importance of using a reporter system with two DSBs in the first paragraph of the Results: "Thus, we posited that a reporter assay that specifically measures EJ of blunt DSBs without causing insertion/deletion mutations (indels) would be specific for C-NHEJ. For this approach, we need to examine EJ between two DSBs (i.e., distal EJ), since repair of a single DSB by EJ without indels restores the original sequence, and hence is not distinct from a site that was never cleaved." Second, we have added the following sentence to the Discussion (paragraph 4). "Although, it is also conceivable that XLF has a unique requirement for EJ without indels between two DSBs (i.e., distal EJ)."

6. The authors used both integrated or extrachromosomal GFP reporters to assess the frequency of C-NHEJ and their genetic requirements. In several places, however, the authors should not compare the outcomes from these two different conditions. It is recommended to modify the manuscript to explicitly describe these conditions in details in the text so the conclusions could be deduced only using the comparable assays.

The reviewer notes that we should take care in the Results to distinguish experiments performed with chromosomally integrated reporters vs. extrachromosomal plasmid experiments, and ensure that we are not comparing experiments using the two different conditions. We have made the requested changes. Specifically, we have clarified the condition of all the extrachromosomal plasmid experiments in the manuscript by adding the word "extrachromosomal" to Fig 2b and Supplementary Fig S2a. As well, in the text, we only compare these experiments with each other, and not to chromosomal experiments. Finally, as mentioned in our response to point #1, we have removed all comparisons of chromosomal vs. extrachromosomal reporters from the manuscript.

Reviewer #2:

We invite the authors to address the minor points indicated below.

1) Figure 3B. The authors convincingly demonstrate the necessary function of KU70 and XLF in EJ that utilizes 1-2 bps of microhomology and no effect of KU70 and XLF deficiency on EJ that utilizes 3-4 bps microhomology. They conclude that C-NHEJ involves EJ with less than 3 bp microhomology and Alt-EJ promotes EJ with ≥ 3 bp microhomology (discussion, page 17). To strengthen these findings, it would be important to determine whether deletion of Alt-EJ factors causes a reduction in EJ with ≥ 3 bp microhomology using the EJ assays described in Figure 3A.

The reviewer shares the same point as Reviewer 1 point #2, such that our responses to these concerns are similar. The reviewer is interested in analysis of other factors in our new EJ reporters, particularly factors that have been implicated in Alt-EJ, and thereby provide a contrast with C-NHEJ factors. We have responded with new experiments that show the influence of CtIP and POLQ on our EJ reporters (new Fig 4). From this analysis, we find that CtIP and POLQ are each important for EJ using 4 nts of microhomology that is embedded from the edge of a DSB. In contrast, CtIP depletion causes a modest increase in EJ using 1-2 nts of terminal microhomology, and EJ without indels. Similarly, POLQ-deficiency causes a modest increase in EJ using 1 nt of terminal microhomology, and has no effect on EJ without indels or using 2 nts of terminal microhomology. Accordingly, CtIP and POLQ have distinct effects vs. C-NHEJ factors on these EJ events. Notably, for EJ using 3-4 nts of terminal microhomology, CtIP has no influence, and POLQ is only modestly important. Accordingly, we have edited the Discussion that describes distinctions between Alt-EJ and C-NHEJ (paragraph 3). As an added note, to strengthen the analysis of C-NHEJ factors on these EJ events, we have added experiments with *Xrcc4*^{-/-} mESCs and the microhomology reporters (new Supplementary Fig S2b), and found similar results to KU70 and XLF.

2) Figure 4B. What is the effect of the L115A, K160D, R178Q and 4KA XLF mutants (alone and in combination) in the microhomology-mediated EJ assays described in Figure 3A?

The reviewer is interested in the effect of the XLF mutants on other EJ assays besides the EJ7-GFP reporter for distal EJ without indels. In response, we have examined these mutants in the 2 nts microhomology (2 nts μ HOM) assay described in Fig 3A. We chose this particular microhomology-mediated EJ assay because it is robust and substantially dependent on XLF. From this analysis, we find that XLF mutants have a very similar effect on the 2 nts μ HOM EJ event, as for EJ without indels. Accordingly, these findings support the conclusions of the parent manuscript regarding the role of XLF during C-NHEJ. These results have been added to the new Fig 5c.

3) Figures 2B and 4A. The authors should provide images with longer exposures of the XLF (Figure 2B) and XRCC4 (Figure 4A, IP) blots.

The reviewer requests longer exposures of two blots. 1) In response to the first request, we have added a new XLF blot to Fig 2b with a longer exposure. 2) For the IP analysis with XRCC4 (current Fig 5a), the most sensitive XRCC4 antibody also causes overall background on the film, so a darker exposure does not provide a better detection of the bands. Nevertheless, in response to this concern, we have shown an additional co-IP experiment that confirms that XLF forms a complex with XRCC4, and that this complex is disrupted with the XLF L115A/4KA mutant (new Supplementary Fig S2c).

Reviewer #3:

Overall, the simulations provide interesting insights on the possible effects of the point mutations. Nevertheless, the manuscript needs to include a more thorough discussion of the MD results on all the systems and connection to the experimental results, and better description of the methods. In particular:

-The authors only discuss selected results from the MD simulations for the various mutants. The manuscript would greatly benefit from increasing the scope of analysis for the simulations including:

a) distance analysis between K160D/D'161, Y167/Y'167 for all systems, i.e. what is the effect of the double mutants on the dimer interface? not only allosteric analysis, also include allosteric analysis on the single mutants.

The reviewer suggests increasing the scope of the molecular dynamics (MD) analysis. In response, we now show the data on the K160D/D'161 and Y167/Y'167 distances, the average number of infiltrated waters into the coiled-coil domain, and the strength of the allosteric pipelines for wild-type (WT) XLF and all the mutants. This data is in the new Fig 6, Supplementary Table S1, and Supplementary Figs S4, S5, and S6. For each XLF system, we performed 8 MD simulations with different initial velocities and realized that for one system four of the eight replicates were duplicated because they were all started at the same clock time and hence the random seed used for initiating velocities were identical (this is how Amber generates the random seed). To keep the number of samples uniform for all systems we considered four replicates for all the systems for MD analysis. We calculated the average RMSD variation with time for each of the four replicates as shown in Supplementary Fig S5. Regarding allosteric pipeline analysis, we have now calculated the strength of the top ten scoring allosteric pipelines passing from the head domain to the coiled domain of the XLF structure for all the mutants and WT XLF. The cumulative allosteric pipeline strength between head domain and coiled-coil region was calculated for each dimer. Details of the statistical analysis for all this data are given in reply to point 1c. Upon single and double mutations in the XLF dimer, we observe changes in (i) the distances between the helices in the coiled-coil domain, (ii) distances between Y167-Y167' and (iii) changes in the strength of allosteric communication pipelines. How these structural changes translate to the observed effects of mutations on the EJ is complex since there are multiple steps leading to the effect on EJ assays (see also our response to point 1e). Additionally, we note that the XLF crystal structure used as a starting point for the MD simulations does not have the disordered carboxy terminus.

b) The authors are encouraged to add residue-wise cross-correlation analysis to complement the analysis of the MD simulations.

In response to this request, we have now performed the residue-wise cross correlation in Supplementary Fig S6. We do not observe any major distinctions between mutants and WT dimers in regards to the DCCM. Regions showing positive cross-correlation in the WT are similarly showing positive cross-correlations in the mutant dimer simulations.

c) provide statistical analysis on all the results, in the present version it is only inferred from Fig. S3a for two of the systems studied.

In response to this request for statistical analysis, we have now plotted the average over 4 replicates and the standard error was calculated for a one-sided t value of 95% confidence level as $2.35 * S.E.M.$ in the calculated data. All the mean data and 95% confidence one-sided t-statistic value are given in Supplementary Table S1. The significance for the difference in strengths of allosteric pipelines compared to WT was calculated using the Wilcoxon Rank Sum Test. The allosteric communication

pipelines from each dimer simulation (4 replicates of 230 ns combined to form one 920 ns ensemble) were analyzed and pipelines connecting the globular head domain to the coiled-coil region were characterized for their strengths. The strengths of the top ten pipelines were compared using the Wilcoxon Rank Sum Test to determine if the distribution of pipeline strengths is significantly different for any of the mutants compared to the WT. We observe that all of the mutants show a significant increase in the median strength of allosteric communication pipelines compared to the WT.

d) include all analysis in a condensed figure (distance, cross-correlation, statistical analysis, etc.) either in the main text or SI. Also include at least a plot of backbone RMSD to show that/if systems are equilibrated and include a mention in the main text in this regard.

We have responded to this request by including the requested information. Specifically, we have now provided all the data and statistical analysis in Supplementary Table S1. The allosteric pipelines data for all the systems is given in Supplementary Fig S4. The moving average of the RMSD plots with respect to the starting structure shows the convergence of the simulations for each system as shown in Supplementary Fig S5. The residue wise cross correlation analysis is shown in Supplementary Fig S6. All this data have been cited either in the Results or the Methods section of the main text.

e) are the combination of distance/allosteric pipeline/cross-correlation results consistent with the experimental results, specifically with respect to the observed effects of single vs. double mutants? i.e., the allosteric analysis for L115A/K160D suggest (not indicate) a possible reason for the observed effect of the double mutant. However, the allosteric analysis by itself on the double mutants and distance analysis for the single mutants as they are reported and discussed in the manuscript do not provide insights on the experimentally observed reduction in activity for the single mutants compared with the (almost) elimination of activity in the double mutant.

The reviewer requests additional discussion on the possible functional implications of the structural perturbations as gleaned from the MD analysis. The MD simulations mostly provide support to the notion that the K160D mutation disrupts the dimer coupling at the coiled-coil interface. We have centered the Results and Discussion section around this conclusion.

We also present evidence for an allosteric pipeline initiating at the region near L115 of one monomer, traversing through K160 and R178 of both monomers. In the parent manuscript, we included a few sentences that the allosteric pipeline linking L115/K160 could contribute to the synergistic loss of function of XLF during C-NHEJ in the double mutant. Based on the reviewers' concern, we recognize that these sentences were unclear/vague. Our explanation for the synergistic loss of function in the double mutants is that disruption of two key binding interfaces (i.e., any two combinations of the interfaces with XRCC4, KU, or at the coiled-coil region of the XLF dimer) could lead to destabilizing the XLF dimer. In contrast, the results from the allosteric communication analysis are only meant to raise the notion that in addition to disrupting a particular binding interface, mutations can also have long-range effects on overall XLF dynamics. Thus, in response, we have 1) removed statements potentially linking the allosteric pipeline results to the synergistic loss of function of the double mutants, 2) clarified that such findings are meant to raise the possibility that mutations can have additional effects apart from disruption of specific binding interfaces, such as long-range effects on protein dynamics. In summary, while the implications of the allosteric pipeline in XLF for C-NHEJ are unclear, we suggest that including this finding will reinforce the notion that individual residues can have long-range effects on XLF protein dynamics.

-The authors are encouraged to use different notation to identify residues from different monomers (see above) to improve clarity. For example: use unprimed to denote residues from one monomer and primed for another, e.g. Y176 for monomer 1 and Y176' for monomer 2.

We thank the reviewer for this suggestion that clarifies the text. In response to this request, we have now used different notations for each of the monomers as suggested by the reviewer.

-There are several details in the methods section that need to be included:

a) number of water molecules (or size of simulation box),

b) minimum distance from edge of protein to edge of solvent box

c) is the non-bonded cutoff the same for real-space for Coulomb and VdW? did the authors use the default parameters for PME for the long-range Coulomb interactions?d) what thermostat/barostat were employed?

e) did the authors use the CPU or GPGPU version of pmemd?

In response to this request, these details are now included in the Methods section in the revised manuscript. We thank the reviewer for these suggestions to improve the description of the approach.

Reviewers' Comments:

Reviewer #1:

Remarks to the Author:

The revised manuscript by Bhargava et al addressed most of the reviewers' comments including mine and is significantly improved than the original version. In particular, the analysis of the role of PolQ and CtIP in end joining events via 4 bp MH underscores genetic and mechanistic difference between two different end joining events that produce the in-del repair types. The authors also cleverly addressed my previous concern to the effect of cleavage efficiency on the outcome of indel end joining at different Cas9 targets by measuring the total end joining associated with deletions of intervening sequences between the two cleavage sites. They then compared the ratio between the total end joining and indel events to demonstrate that MH events are rare and represent small fraction among the total repair.

To help understand this new result, I would still like to know what are the types of events that do produce indels among the total end joining events and the effect of gene deletion such as CtIP, PolQ, XLF on these patterns. The results might be important to deduce the underlying mechanisms of end joining events and the related gene functions.

I also wonder if unlike end joining of a single DSB, two Cas9-induced cleavage events shown here might be independent on end tethering. The premise is also consistent with that XLF is important in ATM deficient cells because ATM is important for end tethering. If true, XLF might be particularly important for inter-chromosomal end joining. It might be interesting to test inter-chromosomal end joining as previously described in Jasin's group.

Lastly, the author needs to explain why the size of hXLF in Fig. 2b is different than endogenous XLF. The presence of 3X flag in the exogenous expression construct might not explain the apparent size difference.

Reviewer #2:

Remarks to the Author:

In this manuscript, Bhargava et al. have satisfactorily addressed the majority of the points raised by this reviewer. The authors now include in their revised manuscript new findings that solidify their previous observations on the role of C-NHEJ factors in the repair of DNA breaks without indels. We therefore recommend this manuscript for publication in Nature Communications.

Reviewer #3:

Remarks to the Author:

all my comments have been addressed

We thank the reviewer for their suggestions to improve the manuscript. We have responded to each request with additions to the text. Each reviewer comment is shown in *italics/green*, and is followed by our response.

To help understand this new result, I would still like to know what are the types of events that do produce indels among the total end joining events and the effect of gene deletion such as CtIP, PolQ, XLF on these patterns. The results might be important to deduce the underlying mechanisms of end joining events and the related gene functions.

I also wonder if unlike end joining of a single DSB, two Cas9-induced cleavage events shown here might be independent on end tethering. The premise is also consistent with that XLF is important in ATM deficient cells because ATM is important for end tethering. If true, XLF might be particularly important for inter-chromosomal end joining. It might be interesting to test inter-chromosomal end joining as previously described in Jasin's group.

The reviewer has requested analysis of end joining (EJ) indel mutation patterns in cells deficient in XLF, POLQ, and CtIP, along with testing the influence of XLF on a chromosomal rearrangement induced by two double-strand breaks (DSBs). We respectfully suggest that such analysis has been performed in prior published studies, such that repeating this analysis for the current manuscript is unlikely to provide new insight. Indeed, a study from our group in 2017 (PNAS 114:728-733. PMID: 28057860) examined the influence of XLF on both of these aspects of EJ. In this study, we developed a reporter for a large chromosomal rearrangement (0.4 megabase pair deletion) between two Cas9-induced DSBs, and examined the frequency of this rearrangement, EJ rearrangement junctions, as well as EJ junctions at a single DSB. From these experiments, we found that XLF-deficiency did not affect the frequency of the rearrangement, but rather affected the EJ junction patterns. For example, loss of XLF caused an increase in the size of deletion mutations for EJ at a single DSB, and for distal EJ. Similarly, the influence of POLQ and CtIP on indel mutation patterns has been published (e.g., PMID: 29079701 for POLQ, PMID: 21131978 for CtIP). Thus, in response, we have made the following text additions.

1. We have added a new paragraph (Discussion #5) to summarize results from our 2017 PNAS paper, as follows: “Along these lines, it is also conceivable that XLF has a unique requirement for EJ without indels between two DSBs (i.e., distal EJ). However, a prior study from our group demonstrated that XLF is dispensable in mESCs for distal EJ *per se*, based on a reporter for a 0.4 megabase pair deletion rearrangement induced by two DSBs²². Furthermore, this study showed that XLF-deficiency affects indel mutation patterns at a single DSB (e.g., caused an increase in the size of deletion mutations)²².”

2. To Discussion paragraph 3, we have mentioned prior studies on EJ indel patterns in cells deficient in CtIP and POLQ, as follows: “These findings are consistent with other reports that POLQ and CtIP are important for EJ events with deletion mutations that use microhomology^{7,49,50}.”

Lastly, the author needs to explain why the size of hXLF in Fig. 2b is different than endogenous XLF. The presence of 3X flag in the exogenous expression construct might not explain the apparent size difference.

The reviewer is concerned that 3xFlag-XLF shows a different migration than endogenous XLF in Fig 2b. 3xFlag-XLF is approximately 36 kilodaltons, whereas XLF is approximately 33 kilodaltons. We expect these molecular weights to be separated on this immunoblot, which involved a relatively long 4-12% gradient gel. Thus, in response, we have added this sentence to the Fig 2b legend “The relative migration of 3xFlag-hXLF vs. endogenous XLF is consistent with the difference in molecular weight.”